



# Multi-tracer study of gas trapping in an East Antarctic ice core

**Kévin Fourteau**[1], **Patricia Martinerie**[1], **Xavier Faïn**[1], **Christoph F. Schaller**[2], **Rebecca J. Tuckwell**[3], **Henning Löwe**[4],
**Laurent Arnaud**[1], **Olivier Magand**[1], **Elizabeth R. Thomas**[3], **Johannes Freitag**[2], **Robert Mulvaney**[3],
**Martin Schneebeli**[4], **and Vladimir Ya. Lipenkov**[5]

[1]Univ. Grenoble Alpes, CNRS, IRD, Grenoble INP, IGE, 38000 Grenoble, France
[2]Alfred Wegener Institute, Helmholtz Centre for Polar and Marine Research, 27568 Bremerhaven, Germany
[3]British Antarctic Survey, Natural Environment Research Council, Cambridge, UK
[4]WSL Institute for Snow and Avalanche Research (SLF), 7260 Davos Dorf, Switzerland
[5]Climate and Environmental Research Laboratory, Arctic and Antarctic Research Institute, St. Petersburg, 199397, Russia

**Correspondence:** Kévin Fourteau (kevin.fourteau@univ-grenoble-alpes.fr)
and Patricia Martinerie (patricia.martinerie@univ-grenoble-alpes.fr)

**Abstract.** TS1 We study a firn and ice core drilled at the new "Lock-In" site in East Antarctica, located 136 km away from Concordia station towards Dumont d'Urville. High-resolution chemical and physical measurements were performed on the core, with a particular focus on the trapping zone of the firn where air bubbles are formed. We measured the air content in the ice, closed and open porous volumes in the firn, firn density, firn liquid conductivity, major ion concentrations, and methane concentrations in the ice. The closed and open porosity volumes of firn samples were obtained using the two independent methods of pycnometry and tomography, which yield similar results. The measured increase in the closed porosity with density is used to estimate the air content trapped in the ice with the aid of a simple gas-trapping model. Results show a discrepancy, with the model trapping too much air. Experimental errors have been considered but do not explain the discrepancy between the model and the observations. The model and data can be reconciled with the introduction of a reduced compression of the closed porosity compared to the open porosity. Yet, it is not clear if this limited compression of closed pores is the actual mechanism responsible for the low amount of air in the ice. High-resolution density measurements reveal the presence of strong layering, manifesting itself as centimeter-scale variations. Despite this heterogeneous stratification, all layers, including the ones that are especially dense or less dense compared to their surroundings, display similar pore morphology and closed porosity as a function of density. This implies that all layers close in a similar way, even though some close in advance or later compared to the bulk firn. Investigation of the chemistry data suggests that in the trapping zone, the observed stratification is partly related to the presence of chemical impurities.

## 1 Introduction

Deep ice cores drilled in polar ice sheets are climatic archives of first importance in the field of paleoclimatology (e.g., Masson-Delmotte et al., 2013). Indeed, ice cores have the unparalleled characteristic of containing various past climatic information in the ice matrix and the air bubbles within (e.g., Petit et al., 1999; Jouzel et al., 2007; Brook and Buizert, 2018). In particular, the bubbles enclosed in polar ice contain air that dates back to their time of formation, and their analysis can be used to reconstruct the atmospheric composition history over more than 800 kyr (Stauffer et al., 1985; Loulergue et al., 2008; Lüthi et al., 2008; Bereiter et al., 2014; Tison et al., 2015). The bubbles become progressively more enclosed in the ice at depths approximately ranging from 50 to 120 m below the surface of the ice sheet, depending on the local temperature and accumulation conditions. These depths correspond to the transformation of the firn, a name for dense and compacted snow, into fully closed ice that encapsulates bubbles.

The firn column is characterized by the increase in its density with depth, as the material gets compressed by the load pressure of the newly accumulated snow (e.g., Arnaud et al., 2000; Salamatin et al., 2009). As the firn density increases, the interstitial pore network is reduced. Eventually, some sections become small enough to prevent gas movement and pinch, enclosing part of the interstitial air (Schwander et al., 1993). This marks the beginning of the trapping zone where bubbles close and encapsulate atmospheric air. The trapping zone usually spans over about 10 m (Schwander et al., 1993).

The process of gas trapping has impacts on the interpretation of ice core gas records. First, since bubbles are trapped deep in the firn, the enclosed air is always younger than the surrounding ice (e.g., Schwander and Stauffer, 1984). This age difference between the ice and the air is known as $\Delta$age in the ice core community, and it needs to be taken into account to synchronize the measurements performed in the bubbles, such as greenhouse gas concentrations, and the ones performed in the ice, such as temperature reconstruction (Shakun et al., 2012; Parrenin et al., 2013). Secondly, due to the progressive closure of bubbles, an ice layer does not contain air corresponding to a single date, but rather from a wide distribution of ages (e.g., Buizert et al., 2012; Witrant et al., 2012). These distributions act as a moving average on the recorded gas signal, and they attenuate the variability in the record compared to the true variability in the atmosphere (Spahni et al., 2003; Joos and Spahni, 2008; Köhler et al., 2011; Fourteau et al., 2017). Finally, due to the heterogeneous stratification of the firn at the centimeter scale, some layers might close in advance or late compared to their surroundings (Etheridge et al., 1992; Mitchell et al., 2015). The gases enclosed in these layers are thus older or younger than the gases in adjacent layers and appear as anomalous values in the record (Rhodes et al., 2016; Fourteau et al., 2017; Jang et al., 2019). Several studies have investigated the closing of porosity and increase in density in firns from both Greenland and Antarctica (Martinerie et al., 1992; Schwander et al., 1993; Trudinger et al., 1997; Fujita et al., 2014; Gregory et al., 2014) and linked it with the trapping of gas in polar ice (Rommelaere et al., 1997; Goujon et al., 2003; Schaller et al., 2017). One of the major results is the observation of a strong and heterogeneous stratification in polar firn and its correlation with ionic and impurity content (Hörhold et al., 2012; Freitag et al., 2013; Fujita et al., 2016). Moreover some studies have highlighted the direct relationship between the density of a firn layer and the volume of closed pores (Schwander and Stauffer, 1984; Martinerie et al., 1992; Mitchell et al., 2015).

Among the polar regions, East Antarctica is of particular interest for ice core drilling as this is where the oldest ice is retrieved. There is currently an effort to retrieve an ice core that is more than a million years old on the East Antarctic plateau (Fischer et al., 2013; Parrenin et al., 2017; Van Liefferinge et al., 2018; Zhao et al., 2018; Sutter et al., 2019). While several studies focus on the cold and arid sites of this region, none of them to our knowledge attempt to encompass a detailed description of stratification, closed porosity, and gas trapping within the same firn. As particular examples, the study of Fujita et al. (2016) describes the link between density variability and ionic content in three firn cores near Dome Fuji without addressing pore closure, while the studies of Schaller et al. (2017) and Burr et al. (2018) propose detailed descriptions of the pore network closure but without discussing stratification.

Our study thus aims to provide a detailed description of chemical and physical properties along the trapping zone of an Antarctic firn core. This core was drilled at the site of "Lock-In" in East Antarctica. It was chosen as its temperature is similar to the cold sites of Dome C and Vostok where deep ice cores have been drilled. Nonetheless, Lock-In displays a higher annual precipitation rate than Dome C and Vostok. Where possible, the measurements were performed with a resolution allowing the observation of stratification and variabilities at the centimeter scale in the firn. Using high-resolution datasets, we aim at investigating the impact of firn stratification on gas trapping.

## 2 Methods

### 2.1 Lock-In study site

The ice core studied in this article was drilled in January 2016 at a site called Lock-In, located on the East Antarctic plateau, 136 km away from Dome C along the traverse road joining the Concordia and Dumont d'Urville stations (coordinates 74°08.310′ S, 126°09.510′ E). As an East Antarctic plateau site, Lock-In exhibits a fairly low mean annual temperature of $-53.15$ °C (measured at 20 m depth in the borehole), slightly higher than at the deep drilling sites of Dome C and Vostok. The site has an elevation of 3209 m above sea level, and the surface atmospheric pressure was estimated to be 645 mbar using the value at Dome C of 643 mbar (data from the Automatic Weather Station Project) and a pressure elevation gradient of 0.084 mbar m$^{-1}$ (evaluated using pressure differences between D80 and Dome C). Lock-In is also characterized by a relatively low accumulation rate of 3.6 cm w.e. yr$^{-1}$. The accumulation rate was evaluated by locating volcanic events with solid conductivity measurements. Hence, Lock-In has a similar temperature but a higher accumulation rate than Vostok and Dome C, which respectively have accumulation rates of 2.2 and 2.5 cm w.e. yr$^{-1}$ (Lipenkov et al., 1997; Gautier et al., 2016).

The drilling operation retrieved ice down to 200 m below the surface. The amount of material was sufficient in the 10 cm diameter core to perform several parallel measurements, which are described below. During the drilling procedure, air was sampled from the firn open porosity along the firn column. The last pumping was performed at 108 m depth, giving an indication of where the firn reaches full clo-

sure and transforms into fully closed ice. Visual observation of the firn core did not reveal any obvious ice layer.

## 2.2 High-resolution density profile

Density is a parameter of particular interest when studying the enclosure of gas in polar ice. Indeed, the densification speed measures the amount of compaction of the firn and the reduction in pore volume. Hence, the density was continuously measured in the Lock-In firn and ice core. The measurements were performed on the whole 10 cm diameter core from 6 down to 131 m below the surface, at the Alfred Wegener Institute, Bremerhaven, Germany. The measurement technique is based on the absorption of gamma rays by the ice phase while traversing the core. The density is then derived from the ray attenuation using Beer's law. The method of measurement is described in detail in Hörhold et al. (2011).

To be applicable, Beer's law requires the estimation of the core diameter $d$ and the ice-specific absorption coefficient $\mu_{\mathrm{ice}}$. The diameter of the core was measured using a caliper for 1 m long sections. However, due to irregularities this value might not be representative of the entire section. Therefore, ice core parts where ice was missing (for instance parts broken during drilling) were removed from the final dataset. Finally, the value of the specific absorption coefficient $\mu_{\mathrm{ice}}$ was calibrated for each new scan using cylinders of pure ice with known diameters. One should note that the results are quite sensitive to the determination of $d$ and $\mu_{\mathrm{ice}}$. Thus, the high-resolution density data will not be discussed in terms of absolute values but rather in terms of centimeter-scale variations, which are better constrained. This measurement method allows us to retrieve density variations at the sub-centimeter scale.

## 2.3 Pycnometry measurements

Pycnometry is an experimental method allowing the measurement of closed and open porosity volumes in a firn sample (Stauffer et al., 1985; Schwander et al., 1993). In this context, a pore is determined as open if it reaches the edge of the sample. The pycnometry experimental setup is composed of two sealed chambers of known volumes $V_1$ and $V_2$, with a pressure gauge connected to the first chamber and a valve allowing the isolation of $V_2$ from $V_1$. The sample is placed in chamber $V_1$, while chamber $V_2$ is isolated and vacuum pumped. Then, the two chambers are connected, resulting in a homogeneous pressure in both chambers lower than the initial pressure in $V_1$. This pressure drop is linked to $V_s$, the volume rendered inaccessible to gases by the sample in the first chamber, by the relation

$$V_{\mathrm{s}} = V_1 - \frac{\mathcal{R}}{1-\mathcal{R}} V_2, \tag{1}$$

where $\mathcal{R} = P'/P$ with $P'$ and $P$ being the pressure value in chamber $V_1$ respectively after and before the dilation. The

application of the equation requires the knowledge of the volumes $V_1$ and $V_2$. The two volumes $V_1$ and $V_2$ were calibrated using stainless-steel balls and bubble-free ice pieces of known volumes. The derivation of Eq. (1) and the description of the calibration procedure are provided in Sect. S1.1 and S1.2 in the Supplement.

Then the closed and open porosity volumes are computed given that

$$
\begin{aligned}
V_{\mathrm{c}} &= V_{\mathrm{s}} - V_{\mathrm{ice}} \\
V_{\mathrm{o}} &= V_{\mathrm{cyl}} - V_{\mathrm{s}} \\
V_{\mathrm{ice}} &= M/\rho_{\mathrm{ice}} ,
\end{aligned}
\tag{2}
$$

where $V_{\mathrm{c}}$ is the closed pore volume, $V_{\mathrm{o}}$ is the open pore volume, $V_{\mathrm{ice}}$ is the volume occupied by the ice phase, $V_{\mathrm{cyl}}$ is the volume of the sample, $M$ is the mass of the sample, and $\rho_{\mathrm{ice}}$ is the density of pure ice. The density of pure ice was taken as $0.918\,\mathrm{g\,cm^{-3}}$ for a recorded temperature of $-8\,^{\circ}\mathrm{C}$ during the measurements (Bader, 1964; Goujon et al., 2003).

A total of 80 samples were taken within the Lock-In core, with depths ranging from 76 to 130 m. The measured samples were cylinders with typical diameters and heights of 4 cm, obtained with a vertical lathe at the Institut des Géosciences de l'Environnement (IGE), Grenoble, France. The radius and height were measured for each sample using a digital caliper in order to estimate the volume of the firn sample $V_{\mathrm{cyl}}$. To estimate the ice volumes $V_{\mathrm{ice}}$, the samples were weighted. Finally, the densities $\rho$ of the samples were computed as $\rho = M/V_{\mathrm{cyl}}$.

The treatment of uncertainties is reported in Sect. S1.3. One of the main results of this uncertainty analysis is the large uncertainties associated with the sample volumes $V_{\mathrm{cyl}}$ due to their sensitivity to the radii measurements. The large uncertainty on $V_{\mathrm{cyl}}$ affects the determination of the density $\rho$ with a typical uncertainty of $0.01\,\mathrm{g\,cm^{-3}}$.

Following the definition of an open pore from a pycnometry point of view, some pores that are closed in the firn (they no longer connect to the atmosphere) might appear open in small samples. For instance, the blue pore in Fig. 1 is closed when taken in the whole firn column, but open when observed in a small cylindrical sample. This opening of closed pores when sampling firn is known as the cut-bubble effect and will be discussed in Sect. 3.1.2.

## 2.4 Tomography measurements

Computed X-ray tomography is a technique gaining traction for the study of firn densification and pore closure (Freitag et al., 2002; Barnola et al., 2004; Freitag et al., 2004; Gregory et al., 2014; Schaller et al., 2017; Burr et al., 2018). Briefly, several 2-D X-ray scans of the samples are performed under different angles. Then, based on the contrast of X-ray absorption between the ice and pore phases, tomography techniques are able to reconstruct a voxelized 3-D model of the scanned samples. A total of 10 samples from the Lock-In firn

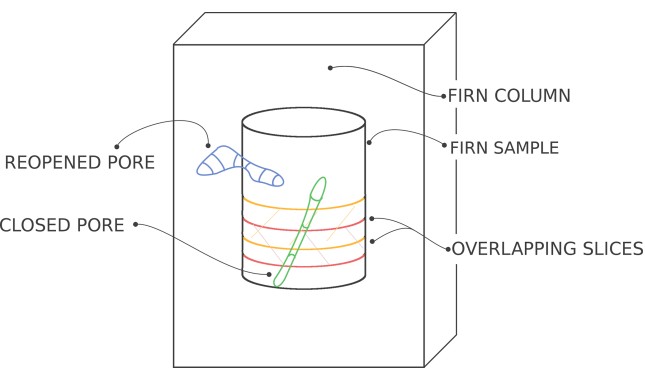

**Figure 1.** Depiction of a cylindrical firn sample, numerical slices in tomographic images, and pores. The yellow and orange slices represent two overlapping slices. The blue pore illustrates the cut-bubble effect: a pore initially closed in the firn column appears open in the cylindrical sample. The green pore is fully enclosed in the cylindrical sample and is thus considered closed in the slices even if it intersects with their boundaries.

core were measured at the Institut für Schnee und Lawinen-forschung (SLF), Davos, Switzerland, using a commercial tomography apparatus. The tomography samples have the same geometry as the pycnometry ones and were also prepared at IGE using a lathe. Four of the samples measured using tomography were also measured by pycnometry. The obtained 3-D models are grayscale images, composed of square voxels of 25 µm in length. The grayscale images were automatically segmented by fitting the gray value histogram to three Gaussian curves (based on Hagenmuller et al., 2013), producing binary images of pore and ice voxels. Using the 3-D images, one can then compute the relative density to pure ice $\rho^R = \rho/\rho_{ice}$ of the samples as the ratio of ice voxels to the total number of voxels. The relative density represents the volume fraction occupied by the ice phase in the firn sample. Throughout the article, we use relative densities instead of densities (mass-to-volume ratio). This was chosen as relative densities are not sensitive to temperature and thus allow an easier comparison between sites.

Tomographic images give access to the pore network and can thus be used to determine the volume of closed and open porosity. As with the pycnometry measurements, a pore is considered open if it reaches the boundary of the sample. Using the software ImageJ and the plug-in Analysis_3D (Boulos et al., 2012), we differentiated the open and closed pores in the 3-D images. The volume of the open and closed porosity is then simply computed by summing the volume of the voxels belonging to an open or closed pore.

## 2.5 Air content

Air content, denoted AC in this article, is a measure of the quantity of air enclosed in polar ice. It is expressed in cubic centimeters of trapped air at standard temperature and pressure (STP) per gram of ice. Air content is directly related to

**Table 1.** Summary of chemistry and impurity measurements, with associated species and typical resolution.

| Method | Resolution (cm) | Depth coverage | Measured species and quantities |
|---|---|---|---|
| Continuous analysis | < 0.1 | 90–115 m | Liquid conductivity |
| IC | ∼ 2.5 | 92–93, 95–96 100–101, and 113–114 m | $SO_4^{2-}$, $NO_3^-$, $Cl^-$, $Na^+$, $Mg^{2+}$, $Ca^{2+}$ |

the volume of pores during closure and can thus be used to estimate the density at which a particular firn layer closed. Air content was measured in the Lock-In firn core using the method described in Lipenkov et al. (1995). In order to correct the AC values for the cut-bubble effect, an estimation of the reopened volume has been performed (based on bubble size measurements; Martinerie et al., 1990). In total 10 samples were measured, including four replicates. Six of these samples were taken around 122 m depth, with a specific focus on stratification, and the four others were taken deeper at 145 m. Based on synchronization between the Lock-In and WAIS Divide ice core methane measurements (Mitchell et al., 2013), the gas ages at 122 and 145 m have been respectively estimated to be 1500 and 1000 CE.

## 2.6 Chemistry along the trapping zone

Several studies highlight the link between density heterogeneities and ionic content in firn layers (Hörhold et al., 2012; Freitag et al., 2013; Fujita et al., 2016). To study the presence of this link in the trapping zone of the Lock-In firn, chemical analyses were performed on this part of the core. The measurements were performed at the British Antarctic Survey, Cambridge, United Kingdom, using a continuous-flow analysis (CFA) system. Sticks were cut from the center of the core (34 mm × 34 mm), covering depths from 90 to 115 m. The sticks were then continuously melted, and the meltwater analyzed for conductivity. Additional meltwater was collected in vials, using a fraction collector, for later analysis with ion chromatography (IC). Four 1 m sections were analyzed at 2.5 cm resolution using a Dionex reagent-free ion chromatography system in a class 100 clean room. The measured species, associated resolutions, and depth coverages are summarized in Table 1.

## 2.7 Continuous methane measurements

During the drilling operation, about 100 m of mature ice was retrieved and analyzed using gas continuous-flow analysis (gas CFA, first developed by Stowasser et al., 2012; Chappellaz et al., 2013). Methane concentration in enclosed bubbles was measured using the gas CFA system

of IGE coupled with a laser spectrometer SARA based on optical-feedback cavity-enhanced absorption spectroscopy (OF-CEAS; Morville et al., 2005). Ice cores were cut in 34 mm by 34 mm sticks, and melted at a mean rate of 3.8 cm min$^{-1}$. The description of this system and data processing is available in Fourteau et al. (2017) and references therein. The continuous measurements were performed from the very bottom of the core ($\sim$ 200 m deep) and melting upward to 100 m below the surface. In the upper part of the core, porous layers are present and let laboratory air enter the measurement system, creating spurious values. The measurements were stopped around 100 m depth, where most of the layers were still porous at the sample scale.

## 2.8 Dataset synchronization

This study produced three high-resolution and continuous datasets: density, chemistry, and methane. Although the chemistry data are composed of various species, they are all based on a unique depth scale and were thus treated as a single dataset for synchronization. In order to produce precise depth scales, notes and measurements were taken when beveled breaks had to be removed from the analyzed ice. Still, uncertainties on the depth scales of the order of a centimeter remain. Since this article will discuss centimeter-scale variability, it is necessary to synchronize the datasets to properly observe the presence or absence of covariation. The synchronization was performed manually, by allowing depth shifts of the order of 1 cm between core breaks.

## 3 Results and discussion

### 3.1 Closed porosity ratios

Deep firn densification is characterized by the transformation of the open pore network into bubbles encapsulating the air. Notably, pore closure is quantified through the evolution of the closed porosity ratio $C_{\text{ratio}}$, defined as the ratio between the closed pore volume and the total pore volume (Mitchell et al., 2015; Burr et al., 2018). This closed porosity ratio falls to zero when the porosity is fully open and reaches 1 when the firn sample porosity is fully closed, making it a simple indicator of pore closure.

For the pycnometry measurements, $C_{\text{ratio}}$ was simply computed for each cylindrical sample from the measured open and closed volumes. On the other hand, thanks to the explicit representation of the pore network, tomography data can be used to access variations at scales finer than the sample size. Hence, the tomography images were numerically divided into 1 cm thick slices, as depicted in Fig. 1. The spacing between two slices was set as 0.5 cm, meaning that slices are overlapping. This overlapping was implemented as an equivalent to a running window analysis. The closed porosity ratio, as well as density were then determined for each slice. To minimize the impact of the cut-bubble effect, the distinc-

tion between closed and open pores was performed on the full cylindrical images as displayed in Fig. 1. Moreover, the top and bottom slices of each sample were discarded since, as described in Sect. S1.4 of the Supplement, they are the most affected by cut bubbles. The values of closed porosity obtained using both pycnometry and tomography are displayed in Fig. 2a against relative density. As pointed out by Mitchell et al. (2015), the relationship between density and closed porosity displayed in Fig. 2 is valid at the centimeter scale, but not necessarily at larger scales.

Burr et al. (2018) proposed using the connectivity index as an indicator of pore closure. This index is defined as the ratio between the volume of the largest pore and the volume of all pores in a given sample and thus equals 1 for a fully open sample and sharply drops near zero for partially closed samples. Burr et al. (2018) report that it is less sensitive to the cut-bubble effect. However, this index cannot be used in models of gas transport and trapping as it does not explicitly represent the volume of closed pores. Therefore, we will use the closed porosity ratio rather than the connectivity index as an indicator of closure in this article.

### 3.1.1 Consistency between pycnometry and tomography

The first closed porosity datasets have been obtained using pycnometry or similar methods (Schwander et al., 1993; Trudinger et al., 1997; Goujon et al., 2003; Mitchell et al., 2015). However, the last decade has seen the development of tomography for the study of firn and estimations of porosity closure based on tomographic images (Freitag et al., 2002; Barnola et al., 2004; Gregory et al., 2014; Schaller et al., 2017; Burr et al., 2018). It is thus important to evaluate the consistency between the two types of methods.

A comparison of the results obtained for Lock-In in Fig. 2a indicates an inconsistency between the two datasets, with the tomography data showing a steeper increase in closed porosity with density. To investigate this difference, we focused our attention on four specific cylindrical samples that have been measured with both the pycnometry and tomography methods. Comparison of the densities of these samples confirms a disagreement between the two methods, with systematically higher density values obtained with tomography. While the differences fall within the uncertainty range of the pycnometry measurements, a stochastic error cannot explain the observed systematic discrepancy. To estimate a possible bias of the tomography-based density, four grayscale images were manually re-segmented. Relative deviations in density values were of the order of $9 \times 10^{-4}$, that is to say 0.1 %. These deviations are similar to the uncertainties reported by Burr et al. (2018). Moreover, Burr et al. (2018) estimated the influence of voxel size on the determination of density values. They found that reducing voxel size from 30 to 12 μm does not change the density by more than 0.1 % for relative densities in the range considered here. Therefore, it

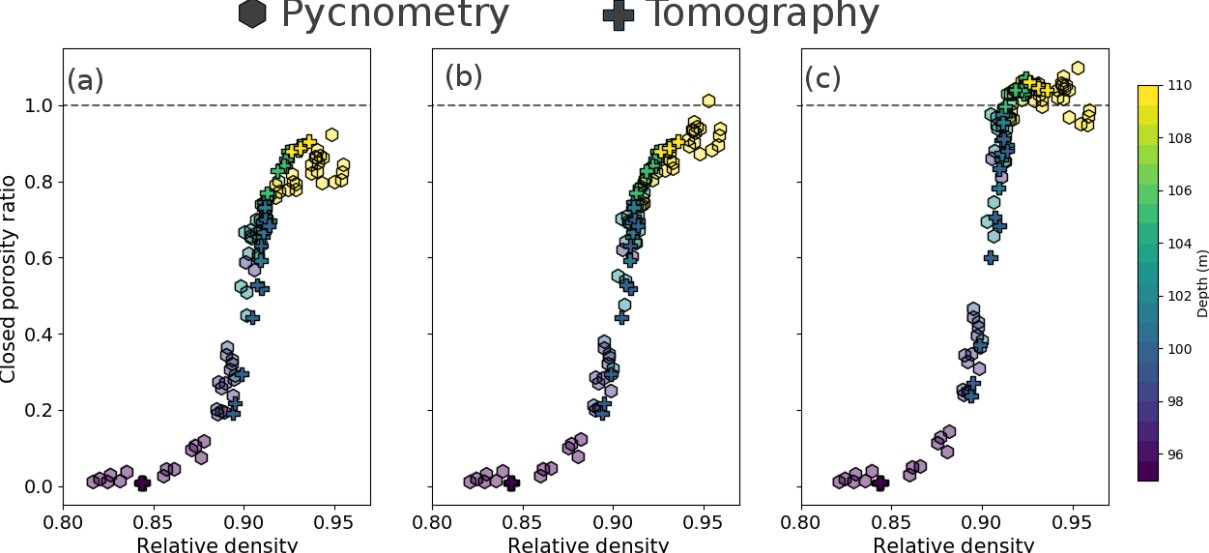

**Figure 2.** Closed porosity ratio against relative density. **(a)** Raw closed porosity ratios obtained from pycnometry and tomography. **(b)** Pycnometry results corrected for volume overestimation and unaltered tomography data. **(c)** Pycnometry and tomography data corrected for cut-bubble effect. Colors stand for the depth of the sample. Note that the color scale has been limited to the range 95 to 110 m to emphasize the middle of the trapping zone.

appears that the densities obtained with tomographic scanning are fairly robust. On the other hand, the densities obtained by size measurements and weighting are sensitive to the volume estimation, particularly through the radius measurement. A close look at the cylindrical samples, notably using the tomography 3-D models, revealed rough surfaces and some broken edges. An overestimation of the measured radius by 0.05 mm is enough to explain the density discrepancy between the four samples measured with both pycnometry and tomography. Our understanding is that while measuring with a caliper, the operator tends to overestimate the actual radius of the sample because of the irregular geometry, leading to a density underestimation. We thus corrected the whole pycnometry dataset by considering a systematic overestimation of the radius by 0.05 mm. This is consistent with the observation of sample surfaces using tomography models, reduces the overall discrepancy between the two methods, and is within the uncertainty range for radius measurements. This radius correction has an influence on the density values but also on the open pore volumes (Eq. 2). The corrected pycnometry results are displayed in Fig. 2b, along with the unaltered tomography results. This correction is sufficient to explain the discrepancy in density between the two methods, while also improving the agreement on measured closed porosity ratios. Thus, after correction, pycnometric and tomographic methods produce consistent results of closed porosity.

### 3.1.2 Cut-bubble effect

As mentioned in Sect. 2.3 and 2.4, during pycnometry or tomography measurements, a pore is considered open if it reaches the edge of the sample. However, some pores that are closed in the firn (that is to say not reaching the atmosphere), are considered open if the edges of the sample intersect with them. This leads to the cut-bubble effect, where a fraction of the closed bubbles are considered open pores (Martinerie et al., 1990; Lipenkov et al., 1997; Schaller et al., 2017). The impact of the cut-bubble effect is clearly visible in Fig. 2a and b: while closed porosity ratio should reach 100 % for high densities (corresponding to a fully closed material), an upper limit of $C_{\mathrm{ratio}} = 90\%$ appears in the data. This is due to the presence of cut bubbles at the surface of the cylindrical samples, even in fully closed ice. This cut-bubble effect can be partially mitigated by the use of large samples.

To correct for this effect, one has to estimate the fraction $f_{\mathrm{reopen}}$ of bubble volume opened during sampling. Then, the true close porosity ratio $C_{\mathrm{ratio}}^{\mathrm{true}}$ is given by

$$C_{\mathrm{ratio}}^{\mathrm{true}} = \frac{C_{\mathrm{ratio}}}{1 - f_{\mathrm{reopen}}}. \qquad (3)$$

Martinerie et al. (1990) report a fraction of cut bubbles up to 10 % for deep ice samples nearly the same size as the sample used in this article. However, this value should not be used to correct for cut-bubble effect in the trapping zone. Indeed, this 10 % value was obtained in deep ice where bubbles tend to have cylindrical or spherical shapes. As pointed out by Schaller et al. (2017), closed pores tend to have elongated

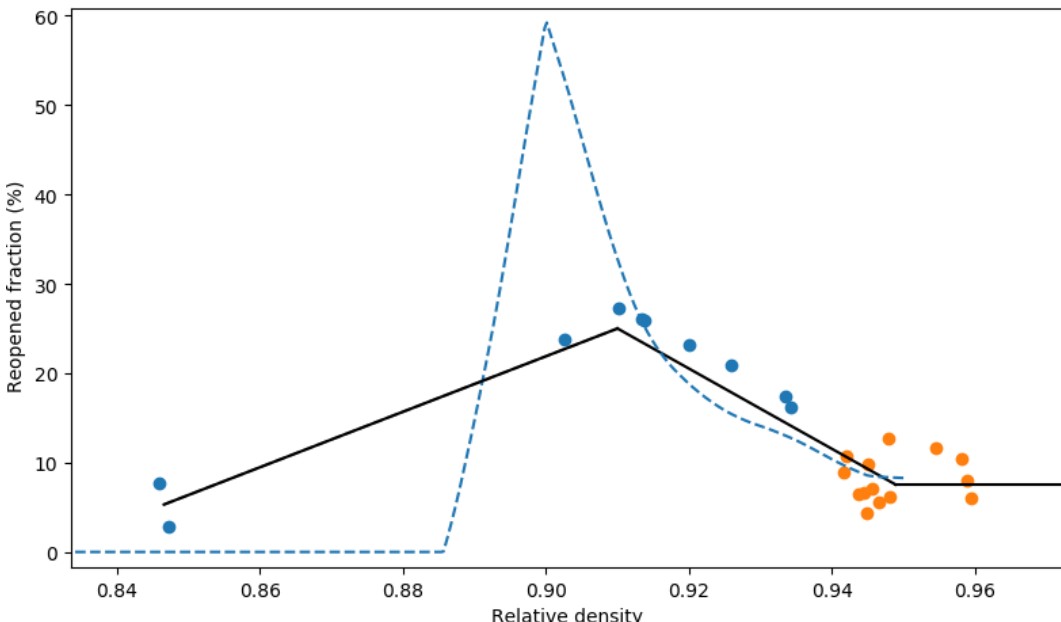

**Figure 3.** Fraction of reopened bubbles as a function of relative density. Values obtained from tomography samples are represented as blue dots, and values obtained from pycnometry measurements are represented as orange dots. The piecewise linear model used to correct for cut-bubble effect is displayed as a solid black line. The correction necessary to reproduce the closed porosities observed by Schaller et al. (2017) is displayed as a dashed blue line.

and tortuous geometries in the trapping zone. The opening of a single pore reaching far into the sample can reopen a significant portion of the closed porosity. They estimated the impact of cut-bubble effect for cylindrical samples of 5 cm in diameter and height, taken from the East Antarctic firn of the site B53 ($-55\,^\circ$C temperature and $3.0\,\mathrm{cm\,we.yr^{-1}}$ accumulation rate). Using tomography, they scanned large firn samples of 10 cm in diameter and 5 cm in height. They then extracted 5 cm diameter cylinders from the large samples. By comparing the state of pores before and after the extraction, they were able to determine which bubbles have been opened. Their findings indicate that up to 60 % of the closed porosity gets reopened near the density of full closure. Moreover, they find that still 30 % of bubbles are reopened when measuring deep ice samples with relative density close to 0.94 (Fig. 4 in Schaller et al., 2017).

However, when applied to the Lock-In closed porosity ratio, the correction proposed by Schaller et al. (2017) leads to unphysical results. For high relative densities, direct application of their correction yields a closed porosity ratio well over 100 %. This indicates an overestimation of the fraction of reopen bubbles for the Lock-In firn.

Two methods were used to estimate the fraction of cut bubbles specifically for the Lock-In samples. First, for depths well below the firn–ice transition, one knows that the samples are fully closed. Using pycnometry, we thus measured closed and open pore volumes in deep samples, and we attributed all the open volumes to the cut-bubble effect. However, this only allows access to the correction for fully closed firn. The

second method was used to estimate the correction in partially closed samples. Similarly to Schaller et al. (2017), a tomography image was numerically reshaped into a smaller subsample. The distinction between closed and open pores in the subsample was then performed following the method of Sect. 2.4. Then, by comparing with the full sample image, one is able to measure the fraction of closed pores that have been opened. To produce the subsamples we trimmed the original cylinder to remove 1 cm in height and 0.5 cm in diameter. The goal is to remove most of the immediate boundary effect described in Sect. S1.4 of the Supplement while conserving a size and a geometry close to the full sample. The method would yield more robust results if, similarly to Schaller et al. (2017), larger volumes were reduced to 4 cm diameter cylinders. However, large-volume tomography was not available for the Lock-In firn core. The deduced fraction of reopened bubbles is displayed in Fig. 3 as a function of relative density. Our findings suggests that up to 25 % of the bubble volume is reopened in our samples for relative densities close to 0.90. This value drops to around 7.5 % for high densities.

In order to correct the $C_{\mathrm{ratio}}$ values obtained by both pycnometry and tomography, we derived a linear piecewise relationship relating the relative density of a sample and the cut-bubble fraction $f_{\mathrm{reopen}}$, displayed in black in Fig. 3. This linear law was derived to match the estimated reopen fraction, while still resulting in physically sound closed porosity ratios, i.e., a monotonous increase without ratios well above 100 %. We then applied the correction from Eq. (3) to the

pycnometry and tomography datasets. The cut-bubble corrected values are displayed in Fig. 2c. It is interesting to note that applying the cut-bubble correction leads to a more abrupt transition at $C_{ratio} = 1$. This observation is consistent with the results of Schaller et al. (2017).

Unfortunately, there is a lack of available data for relative densities ranging from 0.86 to 0.89, as no tomography sample falls in this range. To test the possible sensitivity to a bad estimation of the reopen fraction at these densities, we derived a second correction law with a reopen fraction set at about 25 % for all the relative densities below 0.91. This correction and the corrected data are displayed in the Supplement Sect. S1.5. Application of this correction does not significantly change the final closed porosity data, as low closed porosity values are obtained for relative densities below 0.91.

## 3.2 Reconciling air content and closed porosity

The air content measurements in the Lock-In core indicate a value of $0.0915 \pm 0.0017 \, \text{cm}^3 \, \text{g}^{-1}$ of air at standard temperature and pressure per gram of ice based on four samples at 145 m depth and of $0.0874 \pm 0.0013 \, \text{cm}^3 \, \text{g}^{-1}$ based on six samples at 122 m. These values of air content have been corrected for the effect of cut bubbles, with correction factors of 9.2 % and 5.8 % for the samples from 122 and 145 m, respectively. The difference in air content between the two depth ranges cannot be solely attributed to the effect of cut bubbles. Similar low values of air content have been reported in the upper part of a Vostok ice core (Lipenkov et al., 1997) but not observed in Dome C ice cores (Raynaud et al., 2007). This suggests a potential change of local conditions of the Lock-In and Vostok sites during the late Holocene, which did not occur at Dome C.

Lipenkov et al. (1997) report an air content value of $0.0862 \, \text{cm}^3 \, \text{g}^{-1}$ for the Holocene part of Vostok, while Martinerie et al. (1992) report air content values of 0.0865 and $0.0925 \, \text{cm}^3 \, \text{g}^{-1}$ for the sites of Dome C (1970s drilling site) and South Pole. These three sites have respective elevations of 3471, 3240, and 2835 m above sea level. With an elevation of 3209 m and air content values of 0.0874 and $0.0915 \, \text{cm}^3 \, \text{g}^{-1}$, Lock-In is consistent with these reported values.

The air content value can be used to estimate the effective density of the firn during closure. For this purpose, following Raynaud and Lebel (1979) and Martinerie et al. (1990) we use

$$V_i = V \frac{T_i}{P_i} \frac{P_0}{T_0}$$

$$\frac{1}{\rho_c} = V_i + \frac{1}{\rho_{ice}}, \tag{4}$$

where $V_i$ is the effective porous volume at air isolation (expressed in cubic centimeters per gram of ice), $V$ is the air content (in cubic centimeters STP per gram of ice), $T_i$ and $P_i$ are the temperature and pressure of pores at isolation,

$T_0 = 273.15 \, \text{K}$ and $P_0 = 1013 \, \text{mbar}$ are the standard temperature and pressure, and $\rho_c$ is the estimation of the effective density during closure. Note that $V_i$ is not the porous volume when the firn reaches full closure. Indeed, the volumes of the closed pores evolve in the trapping zone due to compression. Therefore, $V_i$ is the sum of the individual pores' volume when they close and is larger than the porous volume at complete closure. This means that $\rho_c$ is smaller than the density at which full closure is reached and should rather be seen as the typical density at which bubbles are formed when the air pressure starts to deviate from the atmospheric value. Finally, $\rho_c$ can be expressed as a relative density $\rho_c^R$ by assuming a density of pure ice at $-53.15 \, °C$ equal to $0.924 \, \text{g cm}^{-3}$ (Bader, 1964; Goujon et al., 2003).

The air content values determined at depths 122 and 145 m, combined with the local temperature and the pressure at the bottom of the firn column in Eq. (4), yield $\rho_c^R$ values of 0.909 and 0.905, respectively. As shown by the dashed lines in Fig. 4, these densities roughly correspond to the middle of the trapping zone. To further analyze the consistency between the air content values and the closed porosity data, we use a gas-trapping model to estimate the air content value. For this purpose we use the gas-trapping model of Rommelaere et al. (1997) (Eqs. 19a and 21a of their article). This model takes as input the surface pressure and temperature of the site, as well as density versus depth and closed porosity data. Using the hypothesis of similar compression between closed and open pores, the model is able to compute $q_{air}^b$, the quantity of air in bubbles expressed in moles of air per cubic centimeter of ice. This quantity can directly be converted to air content ($V$, in cubic centimeters STP per gram of ice) knowing that $V = q_{air}^b \frac{RT_0}{P_0 \rho}$, where $R$ is the ideal gas constant. To apply the model to the Lock-In core, the measured closed porosity ratios have been translated to a smooth porosity curve (in blue in Fig. 4). Moreover, the high-resolution density data were fitted by a polynomial function of degree 5 to produce a smooth density curve. By applying the local closed porosity with the density relationship to the smooth density profile, we obtained a smooth closed porosity profile. Using it in the model leads to an estimated air content value of $0.102 \, \text{cm}^3 \, \text{g}^{-1}$. This is about 10 % higher than the measured values and well outside the uncertainty range of the measurements. Such a discrepancy has also been reported by Mitchell et al. (2015). Mitchell et al. (2015) argue that a better representation of gas trapping would be to apply the local closed porosity and density relationship to the stratified density profile in order to first obtain a high-resolution and stratified closed porosity profile and to subsequently smooth it. However, as detailed in Sect. S2, application of this methodology did not improve the model–data mismatch.

To explain the discrepancy between data and model, we first investigate the impact of the input uncertainties on the modeled air content value. A first source of discrepancy between the modeled and observed air content values could be an underestimation of the densities in the closed porosity–

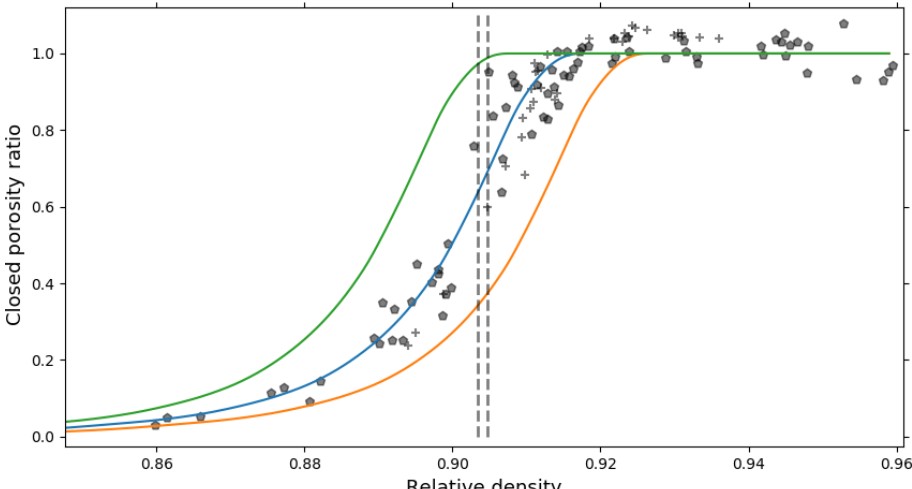

**Figure 4.** Closed porosity law used in the gas-trapping model. Pentagons and crosses are respectively the pycnometry and tomography data (similar to Figure 2c). The blue curve is a fit to these data. The orange and green curves are the closed porosities needed to reproduce the measured air content respectively in the case of iso-compression and incompressible bubbles. The dotted vertical lines represent the relative densities during closure estimated from the two air content values.

density relationship. Indeed, if densities are underestimated, then the model would trap air at too low densities, resulting in a too large bubble volume and too much air. To test this hypothesis, we shifted the smooth closed porosity ratio curve
towards higher densities in order to obtain air content values similar to the ones measured in the ice. To achieve this result, the relative densities have to be shifted by an amount of about $9 \times 10^{-3}$. The resulting closed porosity curve is displayed in orange in Fig. 4. However, such a large shift cannot be rec-
onciled with the pycnometry and tomography measurements. Indeed, the absolute errors on relative densities from the tomography apparatus, estimated as $9 \times 10^{-4}$ in Sect. 3.1.1, are much smaller than $9 \times 10^{-3}$.

A second explanation could be a bad estimation of the
15 cut-bubble correction. Indeed, if we overestimated the cut-bubble effect in the middle of the trapping zone, the model would also trap too much air at low densities and thus over-estimate the air content. To test this hypothesis, we used the extreme case of a cut-bubble correction corresponding to
20 only 7.5 % of opened bubbles throughout the trapping zone. The modeled air content then decreases to $0.096\,\mathrm{cm^3\,g^{-1}}$, still much higher than the measured values of 0.0874 and $0.0915\,\mathrm{cm^3\,g^{-1}}$. A bad estimation of the cut-bubble correction thus cannot explain the model overestimation.
Finally, the discrepancy could be due to a poor estimation of the surface atmospheric pressure and of the temperature at the Lock-In site. Using a surface pressure of 643 mbar, similar to Dome C, and a temperature of $-50\,^\circ\mathrm{C}$, the modeled air content only decreases to $0.0994\,\mathrm{cm^3\,g^{-1}}$. Therefore it
appears that the discrepancy between the modeled and measured air content is not due to errors in the input data of the model.

The discrepancy should thus originate from an incomplete representation of the physical process of gas trapping in the model itself. Schaller et al. (2017) hypothesized a pos-
35 sible impact of stratification on the amount of air trapped. Like previous studies (Stauffer et al., 1985; Martinerie et al., 1992), they argue that the presence of impermeable layers above an open firn layer could retain air in the open porosity. However, this mechanism yields an increase in the amount
of air trapped and can therefore not explain the low value measured in Lock-In ice.

Another explanation could be the wrong representation of closed pore compression in the trapping zone. Originally, the model is based on the iso-compression of bubbles with the
45 rest of the firn; that is to say that bubbles compress at the same rate as the open porosity. However during compression, the pressure inside bubbles increases with the diminution in bubble volume. This potentially makes bubbles less easily compressed than the open porosity. To test the sen-
50 sitivity of the total air content towards bubble compression, we modified Eq. (8) of Rommelaere et al. (1997) to reflect a limited compression of bubbles. For this purpose, we introduce a new parameter $\alpha$ such that when the total porosity decreases by $X$ % during compression, the closed porosity only
55 diminishes by $\alpha X$ %. With this new parameter, the equation governing $q_{\mathrm{air}}^{\mathrm{b}}$, the molar quantity of air in bubbles, is now written

$$\partial_z\left(v q_{\mathrm{air}}^{\mathrm{b}}\right) = -c_{\mathrm{air}} v\left(\partial_z f + \partial_z \epsilon\left(\alpha - 1 - \alpha \frac{f}{\epsilon}\right)\right), \qquad (5)$$

where, similarly to Rommelaere et al. (1997), $z$ is the depth, $v$
is the sinking speed of the firn, $c_{\mathrm{air}}$ is the air concentration in the open porosity expressed in moles per cubic meter of pore space, $\epsilon$ is the total porosity, $f$ is the open porosity, and $\alpha$ is

the parameter representing the degree of compressibility of the bubbles. The detailed derivation of Eq. (5) is described in Sect. S4 of the Supplement. The original Rommelaere et al. (1997) model correspond to the case $\alpha = 1$. The value $\alpha = 0$ represents the case of incompressible closed pores: when a pore reaches closure, it keeps its volume until the end of the trapping process. The intermediate values $0 < \alpha < 1$ corresponds to partially compressible closed pores: they are compressed, but less than the open porosity. Note that this model does not intend to properly represent bubble compression in the trapping zone. Its primary goal is to evaluate the sensitivity of air content to the degree of compressibility of bubbles, and if this mechanism could explain the model–data discrepancy.

Using the model with incompressible bubbles ($\alpha = 0$) yields an air content value of $0.0804\,\mathrm{cm^3\,g^{-1}}$, well below the measured values. Even if the hypothesis of fully incompressible bubbles is not physically supported, this result highlights the sensitivity of the air content to the compression of closed pores in the trapping zone. To yield a value close to the observed air content values in the case of incompressible bubbles, the closed porosity curve needs to have its relative density values shifted by an amount of 0.01 towards lower density. The resulting closed porosity law is displayed in green in Fig. 4. Finally, in order to obtain the measured air contents of 0.0874 and $0.0915\,\mathrm{cm^3\,g^{-1}}$ using the measured porosity data, the $\alpha$ parameter has to be set to 0.35 and 0.50, respectively. This indicates that a smaller rate of compression for the closed pores can reconcile the air content and closed porosity measurements. The optimal $\alpha$ value is sensitive to the cut-bubble correction applied to the closed porosity data. If we use closed porosity data corrected for a maximal reopened volume fraction of 20 %, the $\alpha$ parameter has to be set to close to 0.70 to reproduce the measured air content values. Yet, even if a smaller compressibility of bubbles can reproduce the measured air content, it is not entirely satisfactory as it currently lacks supporting observations. The driver of pore compression in the firn is $\Delta P$, the difference between the overburden pressure of the ice and the air pressure of the pores (Lipenkov et al., 1997). As reported by Martinerie et al. (1992), the $\Delta P$ of open and closed pores differs by less than 8 %, and it is not clear how this $\Delta P$ difference translates in terms of bubble compressibility difference. It is therefore possible that another mechanism is responsible for the discrepancy between the modeled and observed air content. The explanation could be related to the release of gases though capillaries not observable with pycnometry or tomography techniques (Huber et al., 2006; Severinghaus and Battle, 2006). Finally, we also wonder if some bubbles might get reopened in the firn due to the building pressure inside them. Such a reopening of bubbles in the firn would release the gas enclosed at low density and lower the air content value.

The same calculations can be performed for the Vostok firn, where air content (Lipenkov et al., 1997), density (Bréant et al., 2017, and references therein) and pycnometry data (J. M. Barnola TS2, personal communication, TS3) are available. The pycnometry measurements also need to be corrected for the cut-bubble effect. However, contrary to Lock-In we do not have tomographic images of the Vostok trapping zone that could be used to quantify the reopened volumes. Thus, we apply the correction previously derived for Lock-In in Sect. 3.1.2 to the raw Vostok pycnometry data. The application of this correction yields physically sound closed porosity values, with continuously increasing closed porosity ratios up to 1. The uncorrected Vostok data are displayed in Fig. S7. Finally, the Vostok corrected closed porosity data have been fitted to obtain a smoothed closed porosity curve to be used in the gas-trapping model.

Similarly to Lock-In, a direct application of the original Rommelaere et al. (1997) gas-trapping model with iso-compression of bubbles yields a predicted air content of $0.0960\,\mathrm{cm^3\,g^{-1}}$ much higher than the measured value of $0.0862\,\mathrm{cm^3\,g^{-1}}$. This provides further insights that the gas-trapping process might not be well represented in the Rommelaere et al. (1997) model. Again, the introduction of a limited compressibility of the bubbles with a coefficient $\alpha = 0.50$ is able to reconcile the modeled and observed air content values.

## 3.3 Comparison with other sites and porosity laws

The comparison in Fig. 5 of the measured closed porosity data for Lock-In with the proposed parametrization for the B53 site by Schaller et al. (2017) reveals a discrepancy between the two East Antarctic sites. While the B53 closed porosity curve reaches full closure for relative densities of 0.90, the data we obtained for Lock-In reach closure for relative densities closer to 0.92. Therefore, we were not able to reproduce one of their main findings, namely that firn should close at a common critical relative density of 0.90, independently of the studied sites. This discrepancy could be due to an underestimation of the cut-bubble effect for the Lock-In dataset. To test this hypothesis, we derived a new cut-bubble correction that reproduces the closed porosity parametrization proposed by Schaller et al. (2017). As seen in Fig. 3, this new correction assumes a very large reopen fraction around relative densities of 0.90. This is consistent with the results of Schaller et al. (2017) but inconsistent with the reopened fraction of 24 % deduced from the tomography image at a relative density of 0.9025. It is therefore possible that the true Lock-In closed porosities are similar to the ones observed by Schaller et al. (2017) at B53, but a better estimation of the cut-bubble effect is required to answer this question. However, a closed porosity curve similar to the one observed in B53 would trap a higher quantity of air and further increases the discrepancy between the modeled and measured air content described in Sect. 3.2. Even in the extreme case of incompressible bubbles ($\alpha = 0$), the gas trapping used in Sect. 3.2 predicts an air content value of $0.0974\,\mathrm{cm^3\,g^{-1}}$,

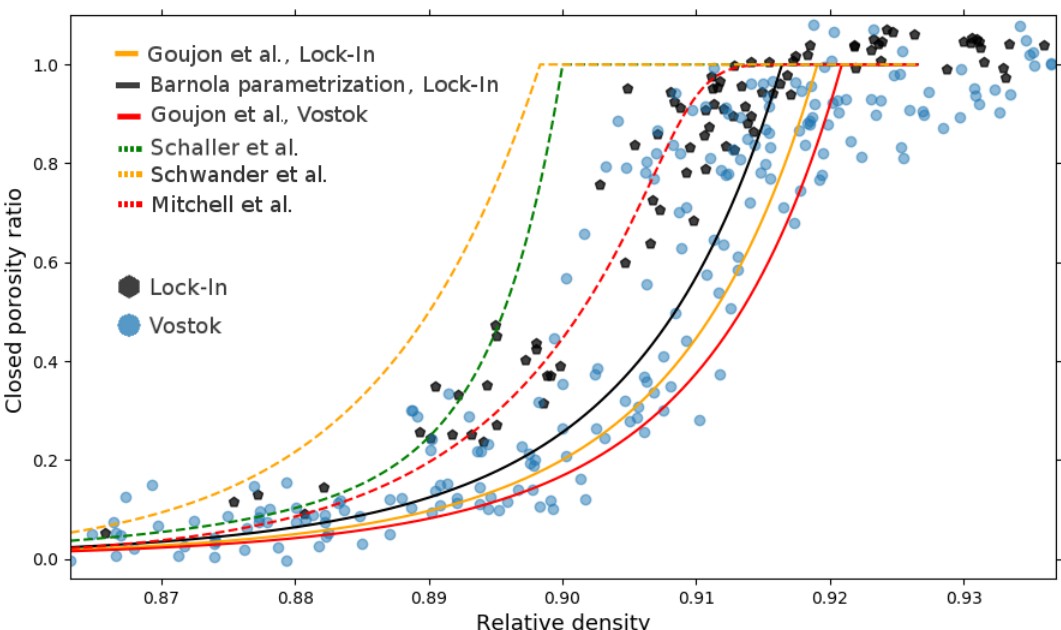

**Figure 5.** Comparison of closed porosity ratio data and parametrizations. The Lock-In closed porosity ratio obtained with pycnometry and tomography is displayed as black hexagons (this study). The closed porosity ratios measured by pycnometry in the Vostok firn are displayed as blue circles (J. M. Barnola TS4, personal communication, TS5). The solid black curve is the Barnola parametrization for Lock-In, using $V_i = 0.114 \, \text{cm}^3 \, \text{g}^{-1}$ (derived from air content data). The orange and red solid curves are the Lock-In and Vostok parametrizations proposed by Goujon et al. (2003) using site temperatures as parameters. The dashed green curve is the parametrization proposed for the B53 site by Schaller et al. (2017). The dashed orange curve is the parametrization of Schwander (1989). The dashed red curve is the parametrization of Mitchell et al. (2015) adjusted to the Lock-In data.

larger than the values of 0.0874 and 0.0915 $\text{cm}^3 \, \text{g}^{-1}$ measured at Lock-In.

Based on previous pycnometry measurements, JM Barnola TS6 proposed an analytical formulation to represent the closed porosity of firn. This parametrization is fully characterized by the effective porous volume $V_i$ (or equivalently the density during closure $\rho_c$) as a single parameter. The derivation of this parametrization is not published in the scientific literature, but Goujon et al. (2003) included this Barnola parametrization in their model (Eq. 9 of Goujon et al., 2003) and used the linear relationship between site temperature and $V_i$ by Martinerie et al. (1994) to model closed porosity data using temperature as the only parameter. Application of the Goujon et al. (2003) parametrization using the Lock-In site temperature yields a predicted effective porous volume $V_i$ of 0.110 $\text{cm}^3 \, \text{g}^{-1}$ and the closed porosity curve displayed in orange in Fig. 5. The resulting closed porosity curve for Lock-In clearly underestimates the closed porosity ratios. In the case of Lock-In, the two air content values can be used to estimate the effective porous volume $V_i$ at 0.109 and 0.114 $\text{cm}^3 \, \text{g}^{-1}$, indicating a possible underestimation of $V_i$ by the temperature relationship. Using the $V_i$ value of 0.114 $\text{cm}^3 \, \text{g}^{-1}$ in the Barnola parametrization results in the closed porosities displayed in black in Fig. 5. This curve is more consistent with the pycnometry and tomography data but still displays a clear underestimation of the closed poros-

ity ratio. Two problems thus arise when using the Goujon et al. (2003) parametrization to predict the Lock-In closed porosities. First, the estimation of the effective porous volume $V_i$ using the temperature relationship might be underestimated, which in turns tends to predict pore closure at too high densities. Then, even with a better constrained $V_i$, the Barnola curve appears to underestimate the closed porosity ratios in the Lock-In firn.

We also displayed in Fig. 5 two other closed porosity parametrizations proposed in the literature, namely the Schwander (1989) and Mitchell et al. (2015) parametrizations. The Schwander (1989) parametrization closes at lower density than the Lock-In data. This is to be expected as this parametrization was proposed to represent relatively warm and high-accumulation sites. The Mitchell et al. (2015) parametrization has been adjusted to fit the Lock-In data ($\rho_{co} = 840 \, \text{kg m}^{-3}$ and $\sigma_{co} = 2 \, \text{kg m}^{-3}$ in Eq. 5 of Mitchell et al., 2015). It shows an overall good agreement with the closed porosity data and results in a 0.099 $\text{cm}^3 \, \text{g}^{-1}$ simulated air content in the Rommelaere et al. (1997) model.

Finally, the closed porosity data measured in a Vostok firn core (J. M. Barnola TS7, personal communication, TS8) are displayed in blue in Fig. 5 alongside the Lock-In data in black. As explained in Sect. 3.2, these data were corrected for cut bubbles using the Lock-In correction derived in Sect. 3.1.2. In the case of Vostok, the Goujon et al. (2003)

parametrization predicts closed porosity ratios (in red in Fig. 5) that are more in line with the data than in the case of Lock-In. Yet, an underestimation of the closed porosity subsists. However, it is possible that this discrepancy between the Goujon et al. (2003) parametrization and the Vostok data is due to a bad estimation of the cut-bubble effect. Comparison of the Lock-In and Vostok data suggests that the Vostok firn reaches closure at a higher density. This is corroborated by air content measurements: the Vostok ice core has a lower effective porous volume at isolation $V_i$, also indicating that the firn reaches closure at a higher density. The difference between the two datasets is more pronounced than the difference between the Lock-In and Vostok Goujon et al. (2003) parametrizations in Fig. 5. Therefore the temperature difference between the two sites does not fully explain the contrast between the two datasets, and it suggests that other climatic parameters such as the accumulation rate or the insolation also influence the porous network and the density range of bubble closure. This is consistent with the work of Burr et al. (2018), who observed different pore network structures in the Lock-In and Dome C firns, two sites with similar temperatures.

### 3.4 Density as a predictor of a firn layer closure

In previous studies, the closed porosity ratio of a firn layer at a given polar site has primarily been linked to its density (Martinerie et al., 1992; Schwander et al., 1993; Mitchell et al., 2015; Burr et al., 2018), despite the firn heterogeneities. This observation is confirmed by our work in the Lock-In firn, where both pycnometry and tomography indicate that the closed porosity ratio of a sample is primarily determined by its density. As depicted Fig. S8 of the Supplement, a larger dispersion is obtained when plotting the closed porosity ratios as a function of depth. In other words, two firn layers with the same density, no matter how far apart they are in the firn column, have nearly the same closed porosity ratio. For example, a particular pycnometry sample taken at a depth of 97.40 m has a relative density of 0.90 and a closed porosity ratio close to 60 %. Such values are usually expected for samples around 102 m depth.

However, the fact that different firn layers exhibit a functional relationship between their density and closed porosity ratio does not imply that they are equivalent in terms of pore structure. One can then wonder if especially high- or low-density layers display similar pore structure as the rest of the firn or if they have a significantly different porous networks. To answer this question we used the tomography 3-D images, which explicitly represent the porous network. Three different pore structure indicators were computed on each of the 1 cm thick slices.

The first indicator is the structure model index or SMI. This index was developed by Hildebrand and Rüegsegger (1997) to analyze the structure of bones. It is traditionally used to quantify the degree of rodness, plateness, and spheriness in 3-D images. In this study the SMI will be used as a simple indicator of pore shape, considering that different pore morphologies should result in different SMIs. As pointed out by Salmon et al. (2015) this index only provides sensible results when used on convex structures. For the Lock-In firn, the SMI was computed in each slice using the BoneJ plug-in within ImageJ (Doube et al., 2010). Results from the plug-in indicated that the porous structure is more than 90 % convex, justifying the use of SMI for firn studies (Gregory et al., 2014; Burr et al., 2018). Nonetheless, the obtained SMI values were sensitive to the resolution of the mesh used to compute the index.

The second indicator is the ratio of pore surface-to-pore volume, denoted S / V here. This surface-to-volume ratio was determined using the method presented in Krol and Löwe (2016), and is based on the determination of the two-point correlation function. The S / V ratio is notably dependent on the size of the pore, as smaller objects tend to have higher surface to volume ratio.

Finally, we estimated the effective diffusivity of the firn samples (Schwander et al., 1988; Fabre et al., 2000; Freitag et al., 2002). The estimation was performed using the open-source application TauFactor (Cooper et al., 2016). Effective diffusivity coefficients were obtained for vertical diffusion between the bottom and top sides of each 1 cm thick slice. However, the resulting values should not be used to model gas diffusion in the firn, as the small size of the samples might not properly capture the large-scale behavior of the firn column (Fabre et al., 2000). Still, the comparison of effective diffusivity between individual samples can be used to highlight similarities or differences in their ability to transport gases. In that sense, our effective diffusivity coefficient is similar to the qualitative measure of permeability (QMP) used by Fujita et al. (2009).

In terms of general trend, the changes of the S / V ratio in Fig. 6 are characterized by an increase with higher densities. This is to be expected as increasing S / V ratio is linked to a decrease in pore size with the compression of the firn. Moreover, our results are consistent with the ones reported by Burr et al. (2018) for the same Lock-In firn. They are also consistent with the S / V values reported by Gregory et al. (2014) for the WAIS Divide and Megadunes sites. Similarly, the decrease in effective diffusivity with density is consistent with the constriction of the pore network, where there are fewer and fewer straight paths traversing the whole sample. Even though our results are not directly comparable to the ones of Freitag et al. (2002) due to the difference in methodology, we also find that the effective diffusivity is related to the open porosity by a power law. However, the exponent found in our case is of 2.9, rather than the 2.1 reported by Freitag et al. (2002). Finally, the evolution of SMI is characterized by an increase in values with density, indicating a transformation of the pore shapes with increasing density. The values displayed in Fig. 6 are consistent with the ones reported by Burr et al. (2018) for the Lock-In firn. Similarly to Burr et al. (2018),

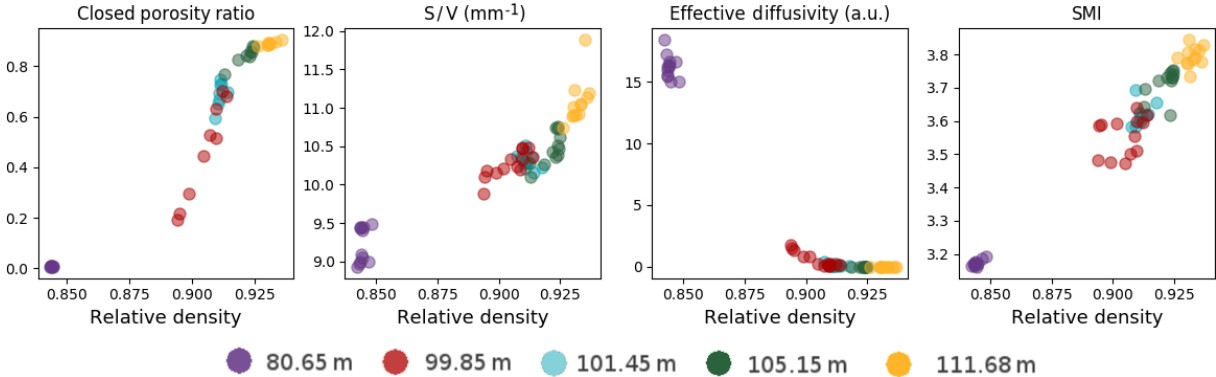

**Figure 6.** Closed porosity ratio and morphology indicators against relative density: closed porosity ratio, S / V ratio, effective diffusivity, and structural model index. Each color corresponds to the depth where the firn samples were taken and covers 10 cm in the firn core.

our SMI values are higher than the ones reported by Gregory et al. (2014) for the WAIS Divide and Megadunes firns. This can be explained by the fact that the SkyScan software used by Gregory et al. (2014) yields lower SMI values (Salmon et al., 2015; Burr et al., 2018).

The tomography data of closed porosity ratio for the firn slices are also reported in Fig. 6. Note that the closed porosity values displayed in Fig. 6 have not been corrected for cut bubbles. It is particularly apparent for the slices taken at the depth of 99.85 m (in red in Fig. 6) that the firn stratification can lead to a wide range of closed porosity ratios within a 10 cm section. Nonetheless, these layers with especially closed or open porosity do not appear as outliers in terms of pore morphology. Their morphological indicators are overall consistent with the other firn samples. Hence, this indicates that all layers tend to follow a similar evolution with density not only in terms of closed porosity ratio, but also in terms of pore morphology and gas transport properties.

### 3.5 Origin of the density variability

We have seen in the previous sections that the degree of closure of a firn layer is primarily a function of its density. However, an important variability of the density is observed at the centimeter scale. One may then wonder about the origin of this density variability in the trapping zone. Previous studies have advanced two mechanisms for the appearance of firn stratification in upper parts of the firn.

The first one is the softening of firn layers due to chemical impurities in the ice phase. Layers with a high concentration of specific ions densify faster, thus creating the density layering. Hörhold et al. (2012) observed a positive correlation between $Ca^{2+}$ and density anomalies in Greenland and Antarctic firn cores. They however do not argue that calcium is the ion specifically responsible for ice softening. Similarly, based on observation in three Dome Fuji firn cores (East Antarctic plateau) Fujita et al. (2016) advanced that the presence of $F^-$ and $Cl^-$ ions facilitates the

densification of a firn layer. They also proposed a hardening effect of $NH_4^+$, preventing the deformation and compaction of firn. Using this hypothesis of ice-softening ions, Freitag et al. (2013) improved the ability of firn models to predict the stratification of deep firn. The second stratification mechanism is detailed in Fujita et al. (2009) and takes into account the ice-phase structure of firn layers. Briefly, due to surface metamorphism, some layers develop bonds between the ice clusters, giving them more resistance to deformation. These layers, which are initially denser at the surface, thus densify slower and are associated with negative density anomalies in deep firn.

Using the high-resolution chemistry and density datasets of the Lock-In firn core we investigate the potential effect of chemistry on deep firn stratification, as proposed by Hörhold et al. (2012), Freitag et al. (2013), and Fujita et al. (2016). The comparison of the high-resolution density and liquid conductivity datasets indicates numerous patterns of covariation between the two signals, with denser layers having higher conductivities. Density and liquid conductivity variations for five 1 m long sections are displayed in Fig. 7, with green and blue portions highlighting some of the observed covariations. This type of covariation is present throughout the whole trapping zone and is consistent with the preferential deformation of firn with high ionic contents. Thus, it appears that the centimeter-scale stratification observed in the trapping zone of Lock-In is in part related to the presence of chemical impurities. Yet, liquid conductivity does not account for all the observed density variability. An illustration can be seen at the depth of 90.30 m (pink section in Fig. 7), where a lower-density layer does not show low values of liquid conductivity. This means one of two things. This less dense layer could be due to the absence of ice-softening impurities, but still contains enough conductive (non-softening) ions for the liquid conductivity not to drop. In this case the density variability is still related to the variability of ice-softening ions but is simply not reflected in the liquid conductivity. Or this less dense layer is not due to a chemistry-

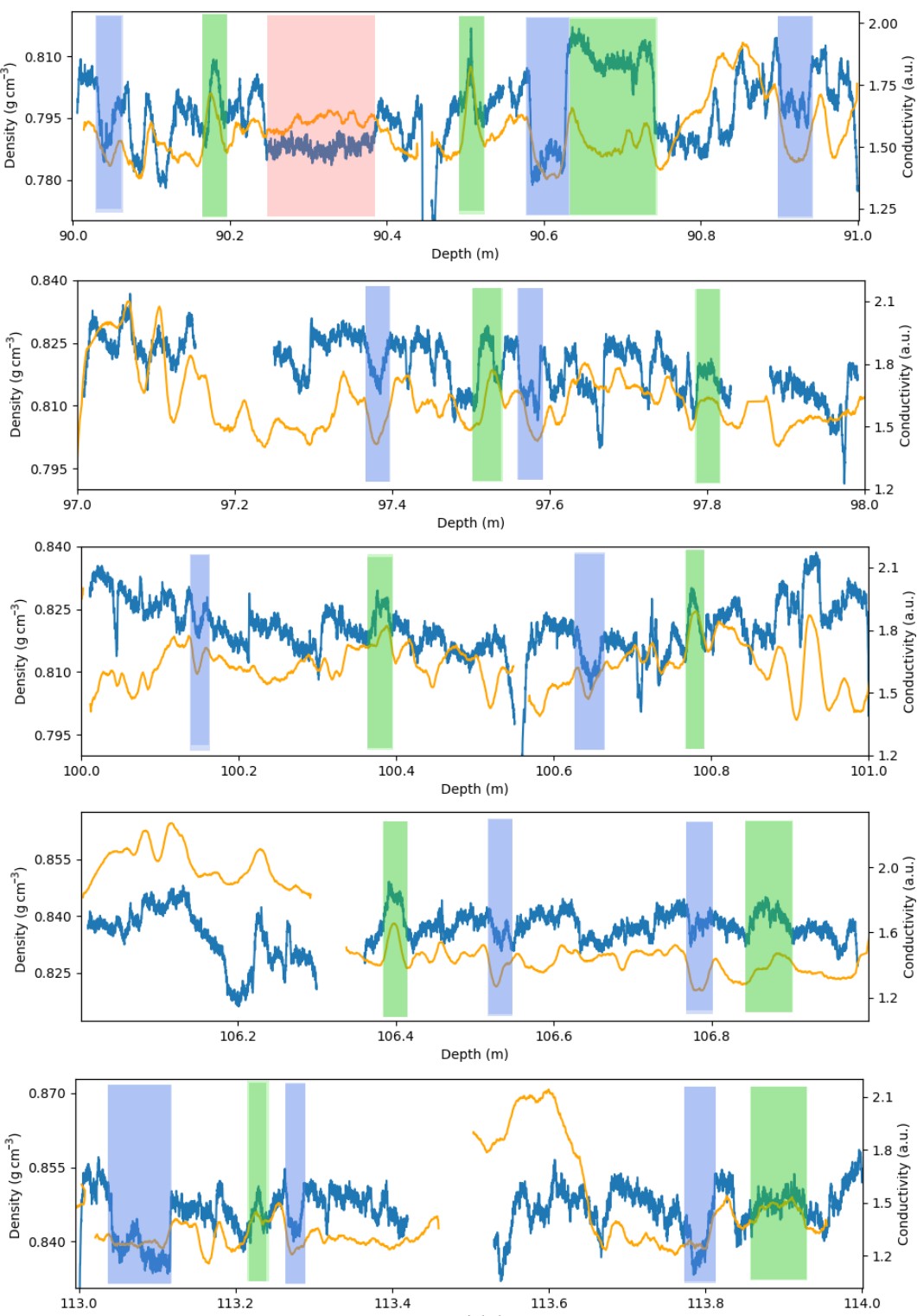

**Figure 7.** Density (blue curve) and liquid conductivity (orange curve) variations in five 1 m long core sections. The green and blue portions respectively highlight some sections where density and liquid conductivity show a simultaneous increase or decrease. The pink portion highlights a section where the decrease in density is not matched by a decrease in liquid conductivity.

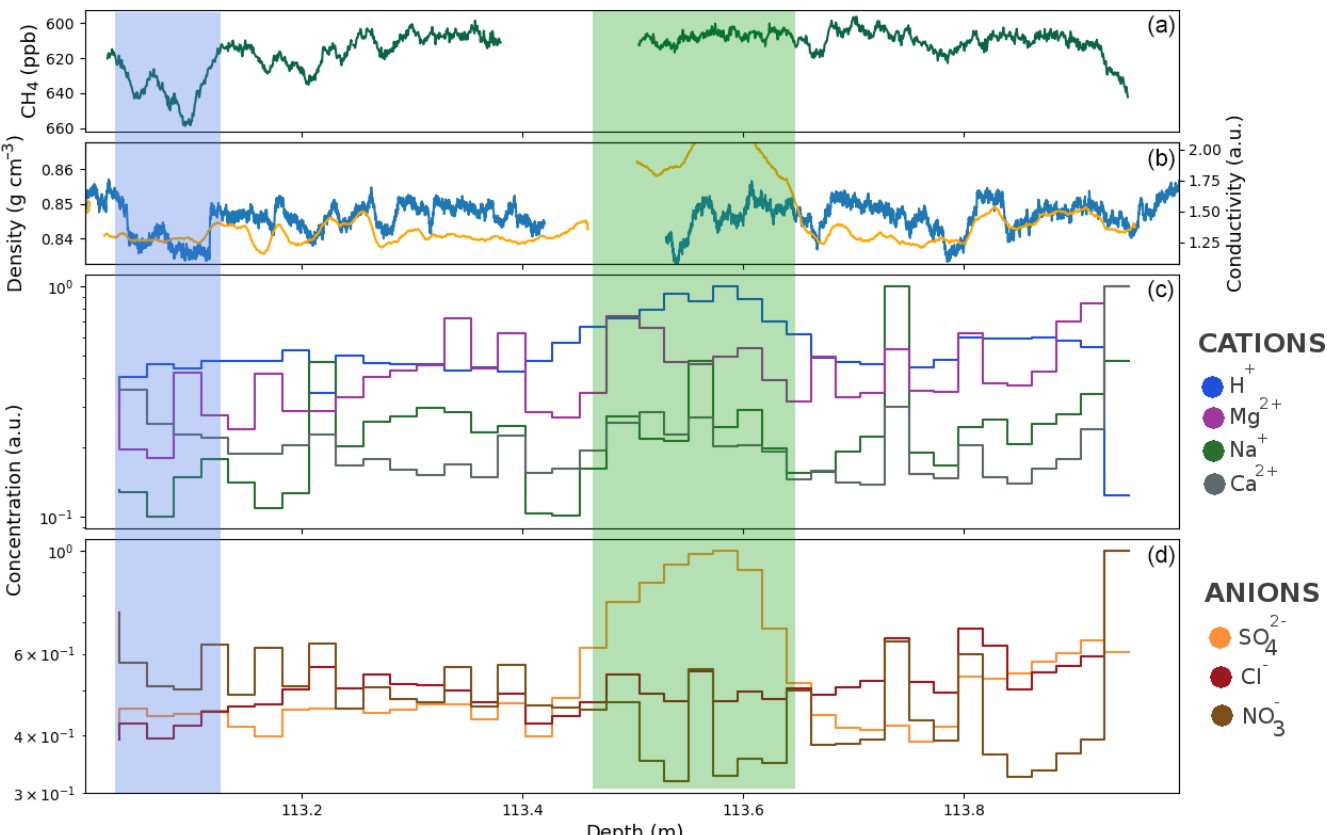

**Figure 8. (a)** Methane concentration measured by CFA between 113 and 114 m depth, with a reversed axis. **(b)** Density (blue) and liquid conductivity (orange) variations. **(c)** Cation concentrations obtained with ion chromatography. **(d)** Anion concentrations obtained with ion chromatography. The ion concentrations are expressed in arbitrary units to highlight their variability. The green portion highlights the presence of a volcanic event. The blue portion highlights a partially open layer.

based effect but rather to another mechanism, for example related to the ice structure as described by Fujita et al. (2009).

In order to better understand the effect of specific ions on the firn density, four 1 m long sections have been measured using ion chromatography with a resolution of 2.5 cm. The concentration of $H^+$ was obtained by closing the ionic balance following Eq. (3) of Legrand and Mayewski (1997). Since $H^+$ ions have a high mobility, $H^+$ concentration estimates could help to explain the centimeter-scale variations in liquid conductivity. The ion concentrations measured on the 113 to 114 m section are displayed in Fig. 8c, d, with density and liquid conductivity variations above. In this section, the data highlight that on one occasion, the sulfate and conductivity undergo strong increases, uncorrelated with other species (green section in Fig. 8). We interpret this spike as a volcanic event, depositing sulfuric acid. It is not associated with any specific effect on the firn density, and we conclude that sulfate has no impact on the layering in the trapping zone. Unfortunately, for other species the results are inconclusive, as we were not able to decipher the effect of specific ions on densification. The same inconclusive observations were made on the three other sections (displayed in

Figs. S11–S13 of the Supplement), where density variations could not be attributed to a particular species measured by ion chromatography. Measurements with a better spatial resolution could potentially help to decipher the impact of specific ion species, similarly to the work of Fujita et al. (2016).

## 3.6 Examples of stratified gas trapping

Thanks to the continuous methane record, we were able to identify layers which were not entirely closed below the bulk firn–ice transition. In the upper part of the methane record, these layers appear as spikes of high concentrations. This is due to the presence of modern air that entered the center CFA stick through the open porosity. An example of a partially open layer is displayed in the blue portion of Fig. 8. This still-open layer is associated with low density and low liquid conductivity values. Please note that the methane axis is reversed in this figure and in the following ones. The same observation was made on the ice core section displayed in Fig. 9. Two late closure layers, depicted by the blue sections, are associated with lower density values and lower liquid conductivities than in the surrounding layers. All these ex-

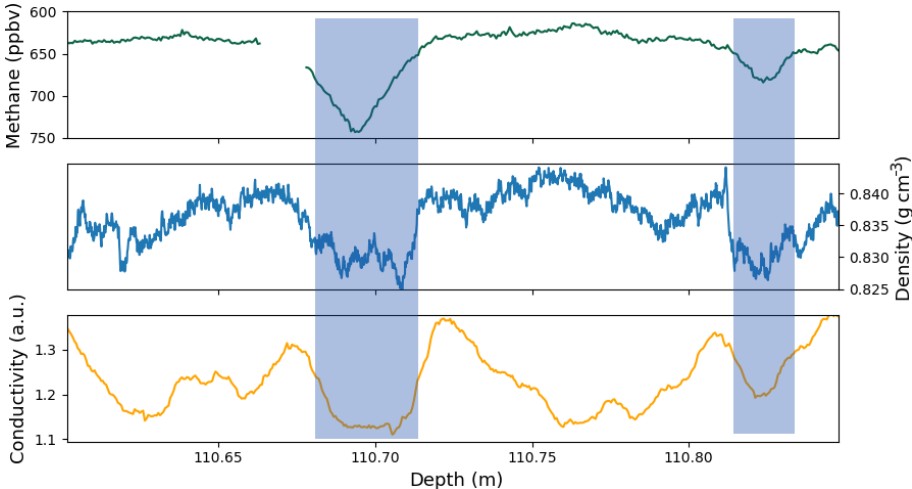

**Figure 9.** Examples of late closure layers highlighted by the blue shadings. The positive spikes in methane around 110.69 and 110.82 m reveal the presence of open pores reaching the middle of the sample (reversed axis). These open layers are associated with low density and liquid conductivity values.

amples are consistent with the previous observation that less dense layers tend to be associated with low ionic contents and thus reach closure deeper in the firn. Figure 9 also reveals a low conductivity layer around 110.77 m not associated with a low density value. This highlights that the presence of potential ice-softening ions is not systematically reflected in the liquid conductivity.

Similarly, an early closure layer was found in the bubbly ice portion of the core (Fig. 10). Following Rhodes et al. (2016) and Fourteau et al. (2017), this layer was detected as a negative methane anomaly during a period of methane rise. This section corresponds to a part of the core with a poor geometry due to a broken edge, and we were not able to properly determine the absolute density values. Still, using the X-ray scan of the unbroken part of the core we could identify the density variability. However, as we do not know the actual thickness of the core that was crossed by the ray, the resulting density variability is expressed in arbitrary units rather than in grams per cubic centimeter. The comparison between density and the methane records in Fig. 10 indicate that the early closure layer corresponds to a denser layer. Air content measurements were performed on this section of the core. Results indicates that the early closure layer does not undergo outlier values of air content. This implies that the early closure layer trapped gases at similar densities as the rest of the firn. It is in line with the description of stratified gas trapping supported in this article, where dense layers close in advance but with the same critical density as others. Unfortunately, no chemical analysis could be performed on this section.

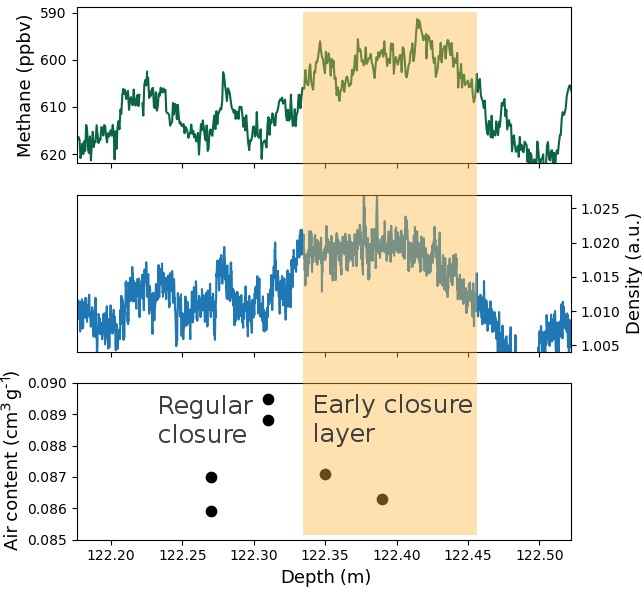

**Figure 10.** Example of an early closure layer highlighted by the orange shade. The reduced methane concentration around 122.40 m is indicative of early gas trapping in the ice (reversed axis). This early closure layer is associated with high density values but air content values similar with the adjacent layer.

## 4 Conclusions

We studied the enclosure of gases at the East Antarctic site of Lock-In, characterized by a temperature of $-53.15\,°C$ and an accumulation rate of $3.6\,cm\,w.e.\,yr^{-1}$. The closed porosity ratio profile in the trapping zone was measured using two independent methods: pycnometry and tomography. Our findings suggest that the two methods yield consistent re-

sults when measuring similar samples and can thus be interpreted in similar ways. However, pycnometry measurements are very sensitive to the determination of sample volumes. On the other hand, tomography appears to provide more robust results and can be used to access more complex characterization of the firn, including pore size and shapes. One important remaining problem in the study of pore closure is the estimation of the cut-bubble effect in the trapping zone as it can drastically change the shape of the closed porosity curve and the estimation of the full closure density. Thus, we encourage the usage of large-volume tomography for future studies of gas trapping and firn densification. Notably, large-volume tomographies of firn where pycnometry data are available could help understand the discrepancy between our results and the results of Schaller et al. (2017).

Using a gas-trapping model, we found that under the assumption of iso-compression of open and closed pores, the measured air content and closed porosity ratios are inconsistent, with the model predicting too much enclosed air. The same overestimation of air content by the gas-trapping model has been found in the Vostok ice core. We have shown that this discrepancy cannot be explained by a poor estimation of the closed porosity data or of the cut-bubble effect. On the other hand, the introduction of a reduced compressibility for closed pores in the gas-trapping model is able to resolve this discrepancy. Yet, this mechanism is not fully satisfactory as it requires a very limited compression of bubbles that has no direct supporting observations. The mechanism responsible for these low quantities of air trapped in ice cores is thus not clear.

Consistently with previous studies conducted at other sites, we observed a strong layering in the Lock-In firn, manifesting itself as centimeter-scale density and firn structure variations (Hörhold et al., 2011; Fujita et al., 2016). Despite this heterogeneity, it appears that all firn layers roughly follow the same behavior relating their closed porosity ratio and pore structure to their density. Layers associated with positive density anomalies close in advance but at the same critical density as others. This vision simplifies the modeling of stratified gas trapping, as it reduces the firn to a stack of equivalent layers.

Finally, using liquid conductivity as a proxy for ionic content, our results suggest that when focusing on the trapping zone, chemical impurities play a significant role in establishing the firn stratification. This does not mean that effects due to the ice-phase structure, as for instance described by Fujita et al. (2009), do not strongly influence firn stratification in upper parts. But, as a first approximation, the effect of chemical impurities appears as the main mechanism for stratified gas trapping. Continuous impurity measurements along the trapping zone with a resolution higher than a centimeter could help decipher the species responsible for the presence of firn heterogeneities.

*Code availability.* The codes used in this study were developed using the Python3 language (with the readily available packages NumPy, matplotlib, SciPy, and skimage), and the macro language of ImageJ. The codes will be provided upon request to the corresponding authors.

*Data availability.* The high-resolution density data in the trapping zone, the pycnometry and tomography closed porosity data, the chemistry data (liquid conductivity and major ions), and the methane record of Lock-In will be made accessible on the World Data Center for Paleoclimatology. The Lock-In smoothed density and closed porosity data used in the gas-trapping model are accessible in the Supplement. The Vostok pycnometry data will be published in a dedicated article; meanwhile, they will be provided upon request to the corresponding authors. CE1

*Supplement.* The supplement related to this article is available online at: https://doi.org/10.5194/tc-13-1-2019-supplement. TS9

*Author contributions.* The high-resolution density measurements were performed by CS, JF, and KF. The chemistry measurements were carried out by RT, RM, LT, and KF. The pycnometry system was restored by OM and LA, and the pycnometry measurements were performed by XF and KF. The tomography scans and segmentations were carried out by HL and MS. The air content measurements were performed by VL. The Lock-In project was supervised by PM and XF. The code development for data processing and modeling was performed by KF and PM. All authors contributed to the discussion and interpretation of the data. The article was written by KF with the help of all co-authors. CE2

*Competing interests.* The authors declare that they have no conflict of interest.

*Acknowledgements.* The Lock-In drilling operation was conducted by Phillipe Possenti, Jérôme Chappellaz, David Colin, Philippe Dordhain, and Patricia Martinerie and supported by the IPEV project no. 1153 and the European Community's Seventh Framework Programme under grant agreement no. 291062 (ERC ICE&LASERS). We thank Sarah Jackson, Emily Ludlow, and Shaun Miller for their important help during the chemistry measurements. We thank Gregory Teste and Grégoire Aufresne for their help cutting and processing the Lock-In core before measurements, Jean-Robert Petit for his help with the solid conductivity data, Anouk Marsal for her help on the tomography data, Samuel Pion for his work on the pycnometry measurements, and Alexis Burr for his help with the ImageJ macro language. We thank Émilie Capron for our useful discussions. We acknowledge the Automatic Weather Station project run by Charles R. Stearns at the University of Wisconsin-Madison, which is funded by the National Science Foundation of the United States of America for providing pressure data at Dome C. This work was supported by the French INSU/CNRS LEFE projects NEVE-CLIMAT and HEPIGANE. We thank Christo Buizert and

the anonymous referee for their constructive comments on the paper. We thank Jean-Louis Tison for editing the article.

*Financial support.* This research has been supported by the NAME OF FUNDER (grant no. GRANT AGREEMENT NO). TS10

*Review statement.* This paper was edited by Jean-Louis Tison and reviewed by Christo Buizert and one anonymous referee.

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

TS2     Please provide full name.

TS3     Please provide year of personal communication.

TS4     Please provide full name.

TS5     Please provide year of personal communication.

TS6     Please provide full name.

TS7     Please provide full name.

TS8     Please provide year of personal communication.

TS9     Please send a new supplement as a *.pdf without the title, authors, correspondence author, etc. as we will generate a supplement title page during publication (with a citation including the DOI), which will contain this information.

TS10     Please note that there is funding information given in the acknowledgements but you have not indicated any funding upon manuscript registration. Therefore, we were not able to complete the financial support statement. Please provide the missing information and double-check your acknowledgements to see whether repeated information can be removed from the acknowledgement. Thanks.

TS11     Please provide date of last access.

TS12     Please provide date of last access.

TS13     Please provide date of last access.