# Peer review of "Multi-tracer study of gas trapping in an East Antarctic ice core"

_The Cryosphere, 2019_

## Referee Comment (RC1) · Anonymous Referee #1 · 9 Aug 2019

This manuscript presents new detailed data on firn densification and gas occlusion for a low accumulation Antarctic site. A wealth of data is presented confirming previous findings in higher resolution and with more certainty. The most important conclusion is that the concept of a critical close off density at which pores are closed off holds. This is important information that potentially conflicts with ideas about total air content variability in the past (e.g. Eicher et al., 2016). The manuscript is well written and I suggest publication with minor corrections (see list below).

P1, Line 20 please write air-bubbles

P1, Line 21 The bubbles contain air from the time of formation. Also rather write that so far the past atmosphere has been measured back to 820kyr BP.

[Figure]

P3, L3 there are plans to retrieve oldest ice in Antarctica based on the site selection from EU, Japan, China, Australia... Cite one of the site selection papers.

P3, L12 "When" should be "Where"

P3, L25 Please give the ball park of the present Vostok and DC accumulation rates.

P16, L20 Write alpha = 0.7 instead of 70%

P16, L23 instead of "8%" write "corresponding to alpha = 0.92"

Figure 6 The color for 80.65m and 101.45m are not very well distinguishable. If possible please chose a different color.

P24, L9 Sentence is unclear

P25, L11 "...not able to properly..."

---

## Referee Comment (RC2) · Christo Buizert (Referee) · 22 Aug 2019

Fourteau et al. present a detailed and data-rich study of the gas trapping process at an East Antarctic site called "Lock-in". This is the first time such a detailed, multi-proxy study has been performed on a core from the East Antarctic plateau, making this a very valuable addition to the firn air literature. The authors come to several important conclusions. Particularly interesting is their observation that several aspects of firn morphology and connectivity are simply linked to local (i.e. small-scale) density, suggesting the latter to be the key controlling parameter independent of the relation of the local density to the bulk density at that depth. The paper is very well written, and easy to follow with good use of figures and references throughout. I am highly supportive of this paper, and think it should be accepted after minor revision.

My main suggestion for improvement concerns the observation that the firn air bubble trapping model of Rommelaere et al. overestimates air content compared to the observations. The authors suggest one solution, namely that closed pores may densify at a lower rate than open pores. As the authors themselves also observe, this explanation seems very unsatisfactory given that the hydrostatic overburden pressure that drives densification is several orders of magnitude larger than the air pressure difference between open and closed pores.

Mitchell et al. 2015 note a similar mismatch at WAIS Divide when using different parameterizations of closed porosity (green and blue curves in their figure 6C), in which case the model overestimates the air content. The stochastic closed bubble parameterization proposed in that study did simulate the correct air content (red curve). Perhaps what Fourteau et al. observe is the same phenomenon?

Mitchell et al. note that the density-closed porosity parameterizations found in the literature (similar to figure 2C in the current paper) are derived using local (i.e. cm-scale) data, and therefore cannot be applied to bulk (i.e. m-scale) density data (their page 2565, last paragraph). Firn models require smooth density and porosity profiles, but the order in which the smoothing is applied matters. Compare the following two scenarios:

Scenario A: Local density record is smoothed to produce bulk density curve. rho-CP parameterization is applied to bulk density curve to obtain bulk CP curve.

Scenario B: rho-CP parameterization is applied to local density record to obtain local (hi-res) CP record. Local CP record is smoothed to produce bulk CP curve.

From reading the text, it appears that Fourteau et al. used scenario A. This is the common approach in the firn air modeling literature, which is strictly speaking incorrect in my view. As Mitchell et al. argue, Scenario B is the correct one. Using scenario B instead of A will reduce the simulated air content, because much of the trapping is shifted to deeper depths. Perhaps the authors did already use approach B, in which
case this point is irrelevant (please clarify in the text). However, the authors are still encouraged to try the stochastic porosity parameterization of Mitchell et al. (their Eq. 5-9) to see if this can reduce the model-data mismatch.

Hopefully this approach can explain the mismatch seen by the authors.

Other comments:

P2 L10: Witrant et al. investigate the thickness of the lock-in zone, which is technically speaking not the same as the trapping zone.

P2 L23: Note that this phenomenon was also recently observed at Styx Glacier by a Korean team (Jang et al., https://doi.org/10.5194/tc-2019-17).

P7L25: consider replacing "airtight" with "fully closed" or "mature"

P8L1-5: Consider citing the original papers on this method, such as Chappllaz et al 2013 and Stowasser et al. 2012

P9L19: consider citing Freitag et al. 2002 here

Figure 2, caption. Perhaps you can emphasize that the rho-CP relationship applies at the cm-scale, and cannot be applied to bulk density data.

Figure 2: Could you add a panel where you compare your new data to the various rho-CP parameterizations that are found in the literature? Barnola, Schwander, Mitchell, etc?

Figure 2: It is very interesting to see that applying the cut-bubble correction correctly (the way you do) makes the transition at CP=1 much sharper/abrupt. That is an interesting finding. The prediction from percolation theory is indeed that the transition should be abrupt – however, previous data sets often made this transition appear somewhat smooth. This difference has long puzzled me, and so the smoothness may have always been an artifact of the cut-bubble effect. Please discuss this briefly. In Mitchell et al. we introduced the parameter sigma_co to fit the smooth transition in the available

porosity data (page 2566); based on your observation, we may have overestimated the value of this parameter (could you suggest a better value?!).

Equation 3,4: it is common in physics to use single (not multiple) letters to denote variables – this avoids confusion when multiplication is involved. Why not use f instead of frac and R in stead of CP, and V instead of AC (or similar). "AC" in an equation makes one think you're multiplying "A" and "C".

P12 L15: "close" instead of "closed"

P13: Can you estimate the gas ages at 122 and 145 m depth, for our estimation?

A dome site like EDC probably has more stable accumulation than a flank site like vostok or lock-in – as the site moves over basal topography the surface slope (and thereby snow redeposition by winds) can change.

P13 L10: estimate the "effective" density of the firn closure

P14L1: the "effective" total porous volume of air isolation

P14 L22: see Mitchell et al. (2015) Figure 4, who also found that models overestimate air content when using standard parameterizations

P14 L18-22: please be explicit about the order of calculating CP from rho and performing the smoothing. So did you use scenario A or B from my example above

P16 L17: Why not use fractions 0.35 and 0.5 rather than percentages? You state alpha is between 0 and 1

P16L23: I agree with this reasoning. Maybe state also that the driving overburden hydrostatic pressure is much much larger than these small differences in bubble air pressure, and therefore alpha=1 is very much expected.

P23 bottom line: Maybe also add Freitag et al. 2013

P27 L3: typo, "to" should be "too"

Data availability: what about the CH4 data?

---

## Author Comment (AC1) · 30 Sep 2019

We are thankful to the referee for their constructive review. A point by point answer to the review is provided in the attached supplement.

Best Regards, Kévin Fourteau on behalf of all co-authors

Please also note the supplement to this comment:
https://www.the-cryosphere-discuss.net/tc-2019-89/tc-2019-89-AC1-supplement.pdf

---

## Author Comment (AC2) · 30 Sep 2019

**RESPONSE TO REFEREE 2, cp-2019-89:**

We are thankful to the referee for their useful and constructive comments.
We will first answer to the comments about the mismatch between air content measurements and modeling. A point by point response to the review is provided after. The text in blue is the text of the review, and the corresponding responses follow in black. The typos pointed-out by the referee will be corrected in the article, and are not specifically addressed in this response.

As pointed out by Mitchell et al 2015 and the referee, there is a difference between applying the density-closed porosity relationship to the a smooth density profile (called scenario A in the review) and applying it to a high-resolution profile before smoothing (scenario B). In the article we used scenario A. As pointed-out later, this will be clearly explained in the text (P14L22).

We tested scenario B and its influence on the modeled air content. For this, we created a heterogeneous high-resolution density profile by applying Gaussian variability to the smooth density profile. The firn was divided into two-centimeter-thick homogeneous layers, and each layer was given a random density anomaly. The Gaussian distribution of density anomalies was parametrized to reproduce the features of the high-resolution density measurements (standard deviation = 7.5kg/m3, see Figure 1 below). We then applied the local density-closed porosity relationship to derive a high-resolution closed porosity profile, and smoothed this closed porosity profile (because the Rommelare et al 1997 calculations requires monotonous profile). As pointed out by Mitchell et al 2015 and the referee this shifts a part of the closure profile towards higher-density (see Figure 2 below). However, using this methodology did not lead to a significant change in the final modeled air content (0.1016 cm3/g instead of the 0.102 cm3/g in the article), and does not explain the model-data mismatch. We also performed a sensitivity analysis: altering the thickness of the homogeneous layers and/or the standard deviation of the density anomalies did not improve the model-data mismatch.

Moreover, we would like to point that if all firn layers close in the same fashion (same density-closed porosity relationship at the centimeter-scale), we should expect similar air content in them, despite their difference in closure depth. Indeed, assuming that sealing effects can be neglected, the bubbles in the various layers form at similar local porosities, and with similar temperature and pressure in the open porosity, meaning that the same amount of air is trapped. This is consistent with our air content measurements of an early closure layer reported in Figure 10 of the article. We should therefore expect scenario A and B to result in similar air content values. In our understanding, this is what explains the low air content variability in the mature ice of WAIS (Figures 3 and 4C of Mitchell et al 2015).

[Figure]

**Figure 1.** In orange: high-resolution density measurements of the "Lock-In" firn core. In blue: synthetic density profile reconstructed from smooth data with added random centimeter-scale variability.

[Figure]

**Figure 2.** Closed porosity ratio profiles. In green: centimeter-scale density-closed porosity relationship used with the smooth density profile (scenario A). In blue: centimeter-scale density-closed porosity relationship used with the synthetic density profile. In orange: smoothed version of the high-resolution closed porosity data in blue (scenario B).

We also tested the closed porosity parametrization proposed by Mitchell et al 2015. The parameters can be adjusted to reproduce the measured closed porosity data, but the air content calculated with the Rommelaere et al 1997 method is then similar to the one the article (0.099 cm3/g instead of 0.102cm3/g). To predict a lower air content value, the closed porosity parametrization would have to be shifted toward higher densities, in a way not consistent with the tomography and pycnometry data (see also the last paragraph P14 of the discussion paper).

The other comments of the reviewer are addressed below. Typos are not specifically addressed but will be corrected in the article.

P2 L10: Witrant et al. investigate the thickness of the lock-in zone, which is technically speaking not the same as the trapping zone.
Indeed, we will provide a more relevant reference (Schwander et al 1993)

P2 L23: Note that this phenomenon was also recently observed at Styx Glacier by a Korean team (Jang et al., https://doi.org/10.5194/tc-2019-17).
We will add the Jang et al 2019 article to the list of layered gas trapping observations.

P8L1-5: Consider citing the original papers on this method, such as Chappllaz et al 2013 and Stowasser et al. 2012
We will modify the article to cite the original CFA papers with *"During the drilling operation, about 100m of mature ice was retrieved and later analyzed using gas continuous flow analysis (gas CFA, first developped by Stowasser et al 2012 and Chappellaz et al 2013). Methane concentration in enclosed bubbles was measured using the gas CFA system of IGE coupled with a laser spectrometer SARA..."*

P9L19: consider citing Freitag et al. 2002 here
We will cite Freitag et al 2002 on using CT to distinguish between open and closed pores.

Figure 2, caption. Perhaps you can emphasize that the rho-CP relationship applies at

the cm-scale, and cannot be applied to bulk density data.
The remark that our data apply at the centimeter-scale will be made in the text of the article P9L9 *"As pointed-out by Mitchell et al 2015, the relationship between density and closed porosity displayed in Figure2 is valid at the centimeter-scale, but not necessarily at larger scales."*

Figure 2: Could you add a panel where you compare your new data to the various rho-CP parameterizations that are found in the literature? Barnola, Schwander, Mitchell, etc?
The Schwander and Mitchell parametrization will be added to Figure 5 alongside the Barnola and Schaller parametrizations. We will also introduce them in the text P19L16 *"We also displayed in Figure 5 two other closed porosity parametrizations proposed in the literature, namely the Schwander et al (1989) and Mitchell et al (2015) parametrizations. The Schwander et al (1989) parametrization closes at lower density than the "Lock-In" data. This is to be expected as this parametrization was proposed to represent relatively warm and high-accumulation sites. The Mitchell et al (2015) parametrization has been adjusted to fit the "Lock-In" data ($\rho\_co$ = 840 kg/m3 and $\sigma\_co$ = 2 kg/m3 in Equations 5 of Mitchell et al , 2015). It shows an overall good agreement with the closed porosity data, and results in a 0.099 cm3/g simulated air content in the Rommelaere et al. (1997) model."*

Figure 2: It is very interesting to see that applying the cut-bubble correction correctly (the way you do) makes the transition at CP=1 much sharper/abrupt. That is an interesting finding. The prediction from percolation theory is indeed that the transition should be abrupt – however, previous data sets often made this transition appear somewhat smooth. This difference has long puzzled me, and so the smoothness may have always been an artifact of the cut-bubble effect. Please discuss this briefly. In Mitchell et al. we introduced the parameter sigma_co to fit the smooth transition in the available porosity data (page 2566); based on your observation, we may have overestimated the value of this parameter (could you suggest a better value?!).
Indeed application of the cut-bubble correction leads to a sharper transition at CP=1. We will point it out in the article (P12L20) *"It is interesting to note that applying the cut-bubble correction leads to a more abrupt transition at CP=1. This observation is consistent with the results of Schaller et al 2017."*
To reproduce our data a parameter $\sigma\_co$ = 2kg/m3 in the Mitchell et al 2015 seems appropriate.

Equation 3,4: it is common in physics to use single (not multiple) letters to denote variables – this avoids confusion when multiplication is involved. Why not use f instead of frac and R in stead of CP, and V instead of AC (or similar). "AC" in an equation makes one think you're multiplying "A" and "C".
We will modify the variable names to be more readable.

P13: Can you estimate the gas ages at 122 and 145 m depth, for our estimation?
We will put the information will be put P7L14, with the description of the air content samples:
*"Based on synchronization between the "Lock-In" and WAIS Divide ice core methane measurements (Mitchell et al 2013), the gas ages at 122 and 145m have been respectively estimated to be 1500CE and 1000CE."*

A dome site like EDC probably has more stable accumulation than a flank site like vostok or lock-in – as the site moves over basal topography the surface slope (and thereby snow redeposition by winds) can change.
It is indeed possible that the variations of air content observed in the Vostok and "Lock-In" ice might be due to changes in accumulation. We however do not have observations to support or reject this hypothesis. Based on radar data (Figure 8 of Verfaillie et al 2012), it appears that "Lock-In" did not experience strong variations of accumulation in the last centuries, but it is not clear that this

observation can extrapolated back in time.

**P14L1: the "effective" total porous volume of air isolation**
We will change the name of Vi to "effective porous volume", which is easier to read than effective total porous volume.

**P14 L22: see Mitchell et al. (2015) Figure 4, who also found that models overestimate air content when using standard parameterizations**
We will add the sentence *"Such a discrepancy has also been reported by Mitchell et al 2015"*.

**P14 L18-22: please be explicit about the order of calculating CP from rho and performing the smoothing. So did you use scenario A or B from my example above**
As explained before, we used scenario A to compute the air content. It will be made clear in the text with the sentence *"By applying the local closed porosity and density relationship to the smooth density profile, we obtained a smooth closed porosity profile. Using it in the model leads to an estimated air content..."*.
Using scenario B only has a minimal effect on the final air content. This will be pointed out P14L22 *"Mitchell et al 2015 argue that a better representation of gas trapping would be to apply the local closed porosity and density relationship to the stratified density profile, in order to first obtain a high-resolution and stratified closed porosity profile, and to subsequently smooth it. However, application of this methodology did not improve the model/data mismatch."*

**P16L23: I agree with this reasoning. Maybe state also that the driving overburden hydrostatic pressure is much much larger than these small differences in bubble air pressure, and therefore alpha=1 is very much expected.**
This point will be made clearer in the text with the sentences *"The driver of pore compression in the firn is $\Delta P$ , the difference between the overburden pressure of the ice and the air pressure of the pores (Lipenkov et al., 1997). As reported by Martinerie et al. (1992), the $\Delta P$ of open and closed pores differ by less than 8%, and it is not clear how this $\Delta P$ difference translates in terms of bubble compressibility difference."* . Due to the non-linearity of ice and the complex micro structure of firn, it is not clear what precise value is expected for alpha.

**P23 bottom line: Maybe also add Freitag et al. 2013**
We will add P22L23 *"Using this softening ions hypothesis, Freitag et al 2013 improved the ability of firn models to predict the stratification of deep firn"*.
We will also add a citation of Freitag et al 2013 at the bottom of P23.

**Data availability: what about the CH4 data?**
The methane data will be made available alongside the rest of the dataset. The data availability section will be modified accordingly.

Best Regards,
Kévin Fourteau on behalf of all co-authors

---

## Author Response (AR1)

**Response to the Editor :**

Dear Editor,

Please find below a version of the article and its supplement, with the differences between it and the previous version of the manuscript highlighted in the text.
Text that as removed is displayed as red strike-out, and text added is displayed as blue underlined.

On behalf of all co-authors,
Kévin Fourteau

[revised manuscript text omitted]

In the main part of the article, the air content of the Lock-In ice core was modeled using the closed porosity versus depth profile obtained by applying the measured closed porosity versus density relationship (Section 3.1) to a smoothed version of the density profile. As pointed-out by Mitchell et al. (2015) , the measured closed porosity curve is obtained with centimeter-scale samples, while the smoothed density data represent a bulk meter-scale profile. The mixing of data at the centimeter and meter scales

10    might be inconsistent and lead to a wrong estimation of the air content. To avoid this issue, Mitchell et al. (2015) propose to apply the closed porosity relationship with density to a centimeter scale density profile, and to subsequently smooth the result in order to obtain the closed porosity versus depth profile needed for the model. We tested whether the approach proposed by Mitchell et al. (2015) could explain the model/data discrepancy for air content.
For this purpose we produce a synthetic high-resolution and stratified density profile by applying a random Gaussian noise to

15    our smooth density profile. We were not able to directly use the high-resolution density measurements, as there are numerous gaps in the data. As displayed in Figure S9, the Gaussian noise was defined with a standard deviation of $7.5\,\mathrm{kg.m^{-3}}$ to reproduce the variability observed in the density data. A high-resolution closed porosity profile was then obtained by applying

[Figure]

**Figure S8.** Closed porosity ratios versus sample depth. Colors stand for the measured relative density.

the centimeter-scale closed porosity versus density relationship. It is displayed in blue in Figure S10. This high-resolution profile was finally smoothed to obtain a continuous and monotonous profile, as required by the Rommelaere et al. (1997) model. This new smoothed porosity profile is shown in orange in Figure S10, and compared with the closed porosity profile used in the main article in green.

5   Using the new closed porosity profile in the Rommelaere et al. (1997) model leads to an air content value of $0.1016\,\mathrm{cm}^3.\mathrm{g}^{-1}$, almost identical to the $0.102\,\mathrm{cm}^3.\mathrm{g}^{-1}$ value of the main part of the article. Therefore, application of the Mitchell et al. (2015) methodology does not explain the model-data discrepancy. We also performed a sensitivity analysis, by modifying the standard deviation of the Gaussian density noise. This did not improve the model results.

10   Moreover, we would like to point that if all firn layers close in the same fashion (same closed porosity versus density relationship at the centimeter-scale), we should expect similar air content in them, despite their difference in closure depth. Indeed, assuming that sealing effects can be neglected, the bubbles in the various layers form at similar local porosities, and with similar temperature and pressure in the open porosity, meaning that the same amount of air is trapped. This is consistent with our air content measurements of an early closure layer reported in Figure 10 of the main part of the article. We should

15   therefore expect the application of the Mitchell et al. (2015) and of our methodologies to result in similar air content values.

[Figure]

**Figure S9.** Comparison of the high-resolution density measurements in orange, and the synthetic density data in blue.

[Figure]

**Figure S10.** Closed porosity ratios versus bulk meter-scale density of Lock-In firn. In blue: high-resolution closed porosity profile obtained by applying the centimeter scale closed porosity versus density relation to the synthetic high-resolution density profile. In orange: Smoothed version of the synthetic high-resolution closed porosity profile. In green: Closed porosity profile used in the main part if the article, obtained by applying the measured the centimeter scale closed porosity versus density relation to the smoothed density profile.

**S3 Ion Chromatography**

In total four one-meter long sections were analyzed using ion chromatography. One of the sections is displayed in Figure 8 of the main article. The three remaining sections are displayed in Figure S11 to S13 with density, liquid conductivity and major ion concentrations data.

[Figure]

**Figure S11.** High resolution density (in blue), liquid conductivity (in orange) and major ion concentrations for the section between 90 and 91 m depth. The ion concentrations are split between anion and cations, and normalized to emphasis variability.

[Figure]

**Figure S12.** Same as Figure S11 for the section between 92 and 93 m depth.

**S4 Gas trapping model**

To model the air content in ice, we used a modified version of the Rommelaere et al. (1997) gas trapping model, introducing a limited compressibility for the closed pores. Here, we describe how the new equation representing air trapping is obtained. The

[Figure]

**Figure S13.** Same as Figure S11 for the section between 100 and 101 m depth.

derivation is similar to the one of Rommelaere et al. (1997), and we only detail the differences with the original calculation. The symbols used and the quantities they represent are the same as in Rommelaere et al. (1997).

Our new model is based on the replacement of Equation 8 of Rommelaere et al. (1997) by:

$$\frac{(\epsilon_1 - f_1) - (\epsilon - f)}{\epsilon - f} = \alpha \frac{\epsilon_1 - \epsilon}{\epsilon} \tag{6}$$

5   where $\epsilon$ and $\epsilon_1$ are the total porosity before and after compression, $f$ and $f_1$ are the open porosity before and after compression, and $\alpha$ is rate of compression of closed bubbles. This equations simply means that during compression, if the total porosity diminishes by $X\%$, the closed porosity diminishes by $\alpha X\%$. The original study by Rommelaere et al. (1997) corresponds to the case $\alpha = 1$.

10   The air conservation in bubbles is expressed by the Equation 13 of Rommelaere et al. (1997). Re-arranging the equations to eliminate the intermediate subscripts 1, and taking the time step $dt$ as infinitely small leads to the equation:

$$d[(\epsilon - f)c_{\text{air}}^{\text{b}}] = -\partial_z v dt(\epsilon - f)c_{\text{air}}^{\text{b}} + (d\epsilon(1 - \alpha + \alpha\frac{f}{\epsilon}) - df)c_{\text{air}} \tag{7}$$

This equation replaces Equation 15 of Rommelaere et al. (1997).

Finally under the assumption of stationarity, this Lagrangian description is converted into an Eulerian one:

15   $$\partial_z(vq_{\text{air}}^{\text{b}}) = -c_{\text{air}}v(\partial_z f + \partial_z \epsilon(\alpha - 1 - \alpha\frac{f}{\epsilon})) \tag{8}$$

The quantity of air trapped in ice $q_{\text{air}}^{\text{b}}$ can then be calculated from this equation. The required inputs for the model are density and closed porosity profiles, to compute the $v$, $f$ and $\epsilon$ variables. For this study Equation 8 was solved using a finite difference

scheme.